# Differentiation and Specialization of Attention Heads via the Refined Local Learning Coefficient

**George Wang**
Timaeus

**Jesse Hoogland**
Timaeus

**Stan van Wingerden**
Timaeus

**Zach Furman**
Timaeus

**Daniel Murfet**
School of Mathematics and Statistics
The University of Melbourne

## Abstract

We introduce refined variants of the Local Learning Coefficient (LLC), a measure of model complexity grounded in singular learning theory, to study the development of internal structure in transformer language models during training. By applying these *refined LLCs* (rLLCs) to individual components of a two-layer attention-only transformer, we gain novel insights into the progressive differentiation and specialization of attention heads. Our methodology reveals how attention heads differentiate into distinct functional roles over the course of training, analyzes the types of data these heads specialize to process, and discovers a previously unidentified multigram circuit. These findings demonstrate that rLLCs provide a principled, quantitative toolkit for *developmental interpretability*, which aims to understand models through their evolution across the learning process. More broadly, this work takes a step towards establishing the correspondence between data distributional structure, geometric properties of the loss landscape, learning dynamics, and emergent computational structures in neural networks.

## 1 Introduction

Structure in the data distribution has long been recognized as central to the development of internal structure in artificial and biological neural networks (Rumelhart et al., 1986; Olshausen & Field, 1996; Rogers & McClelland, 2004). Recent observations have renewed interest in this topic: language models progress through distinct stages of development during training, acquiring increasingly sophisticated linguistic and reasoning abilities in ways that seem to reflect the structure of the data distribution (Olsson et al., 2022; Chen et al., 2024; Belrose et al., 2024; Tigges et al., 2024; Edelman et al., 2024; Hoogland et al., 2024).

A deeper understanding of how structure in the data determines internal structure in trained models requires tools that provide information about *which* components of a model are being shaped in response to *what* structure in the data distribution. Our foundation for the study of such questions begins with the local learning coefficient (LLC; Lau et al. 2023) from singular learning theory (SLT; Watanabe 2009), which is a measure of model complexity. In this paper, we introduce the *refined local learning coefficient* (rLLC), which measures the complexity of a component of the model (for example an attention head or whole layer) with respect to a given data distribution, which may differ from the training distribution. For example, we can measure the rLLC for a particular attention-head in a transformer trained on the Pile (Gao et al., 2020; Xie et al., 2023) with respect to the distribution which conditions on a token sequence representing code.

We focus mainly on the rLLCs of individual attention heads and demonstrate the utility of these metrics in studying the progressive differentiation and specialization of heads. The diversity of attention heads at the end of training has been established in recent years through mechanistic interpretability, which has provided numerous examples of attention heads that appear to have specialized functions, including previous-token heads (Voita et al., 2019; Clark et al., 2019) and

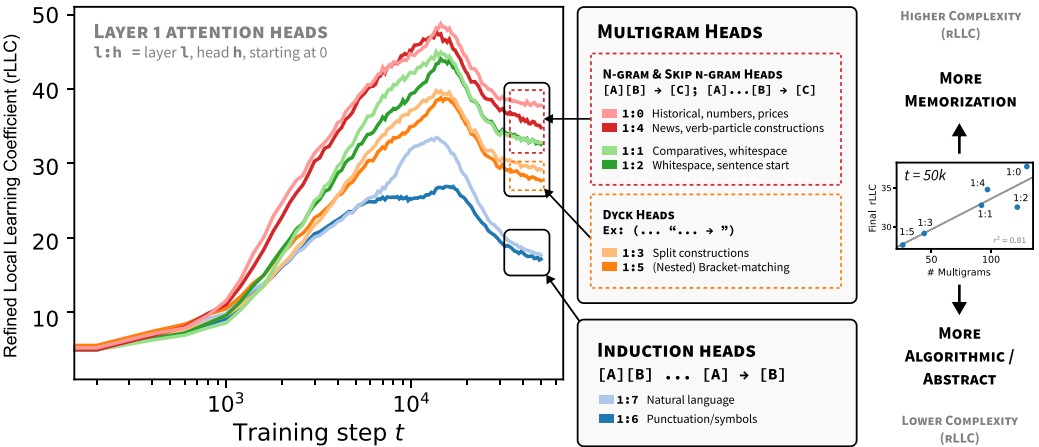

Figure 1: **The weight-refined local learning coefficient (wrLLC) measures the *complexity* of model components (such as attention heads) over training**. At the end of training, heads with lower wrLLC can be described by simple algorithms (e.g., induction heads, bracket-matching), whereas heads with higher wrLLC memorize $n$-grams and skip $n$-grams ("multigrams"). Shown on the left are the wrLLC curves over training for layer 1 heads, automatically clustered by $K$-means (clusters are indicated by a dominant color, within which individual heads are distinguished by shading). The clusters match the head types (middle-right, classified in Appendix B). Final rLLC correlates with the number of memorized multigrams for each multigram head (far-right).

induction heads (Olsson et al., 2022) among other kinds (Wang et al., 2023; Gould et al., 2024). In this paper, we are interested in understanding how this diversity *emerges* over the course of training, in a way that may eventually replace the need for detailed head-by-head mechanistic analysis.

Our methodology yields several novel insights. In the same two-layer attention-only transformers studied in Hoogland et al. (2024) we show that:

- **Weight-refined LLCs reveal how attention heads *differentiate* across training**: although initially the rLLC curves for heads across training are similar, they progressively diverge with distinctive patterns for different types of heads (Figure 1 and Section 4.1).

- **Data-refined LLCs reveal how attention heads *specialize* across training**: a leading hypothesis for the driver of specialization is structure in the data, which imprints order into a neural network. We show that the data rLLC reflects this specialization, for example by showing that one induction head appears partly specialized to code (Section 4.2).

These findings are validated using independent, established techniques, such as clustering algorithms and ablations. We then demonstrate how to use these refined LLCs in combination, which results in our third key contribution:

- **The identification of a novel multigram circuit**: refined LLCs reveal evidence of internal structure related to multigram prediction, which we corroborate with other interpretability techniques (Section 4.3).

The developmental perspective, where we pay attention to the transformer across the entire training process, is crucial to this methodology. The internal structure of mature organisms is constrained by the fact that they have to develop from embryos (Barresi & Gilbert, 2023). Similarly, the fact that the structure of a neural network has to develop from initialization in a way that decreases the loss at each stage should be a source of valuable insight into the trained model, but to date the field of interpretability has not focused on this source of information. We view our results as helping to establish the validity of a new field of developmental interpretability, which emphasizes the interplay between distributional structure in the data, geometric structure of the loss landscape, structure in learning dynamics, and resulting computational structure in the model.

## 2 SETUP

Following Elhage et al. (2021), Olsson et al. (2022), and Hoogland et al. (2024), we study two-layer attention-only (without MLP layers) transformers (architecture and training details in Appendix F) trained on next-token prediction on a subset of the Pile (Gao et al., 2020; Xie et al., 2023). Throughout, we refer to the attention head in layer $l$ with head index $h$ by the notation $\boxed{l\text{:}h}$ (starting at 0). We also contextualize the development of different components of this model by using the same macroscopic stages LM1 through LM5 from Hoogland et al. (2024) as the backdrop for figures (see top of Figure 2). These stages correspond to (LM1) learning bigrams, (LM2) learning $n$-grams and skip $n$-grams ("multigrams"), (LM3) developing previous-token heads, and (LM4) developing induction heads before (LM5) converging.

We denote an input context, a sequence of tokens $t_k$, by $S_K = (t_1, \ldots, t_K)$ where $K$ is the context length. We denote by $S_{\leq k}$ the sub-sequence $(t_1, \ldots, t_k)$ of $S_K$. Our data $D_n$ is a collection of length-$K$ contexts, $\{S_K^i\}_{i=1}^n$, from a data distribution $q$, indexed by the superscript $i$.

The *empirical loss* with respect to $D_n$ is

$$\ell_n(w; q) = -\frac{1}{n} \sum_{i=1}^{n} \frac{1}{K-1} \sum_{k=1}^{K-1} \log\left(\text{softmax}(f_w(S_{\leq k}^i))[t_{k+1}^i]\right), \tag{1}$$

where for a probability distribution $P$ over tokens we denote by $P[t]$ the probability of token $t$, and $f_w$ is the function from contexts to probability distributions of next tokens computed by the transformer. The corresponding *population loss* $\ell(w; q)$ is defined by taking the expectation with respect to the true distribution of contexts. When $q$ is the pretraining distribution, we suppress $q$ and write $\ell_n(w)$.

## 3 METHODOLOGY

The (global) learning coefficient $\lambda$ is the central quantity in singular learning theory (Watanabe, 2009). In this section we review the local learning coefficient (LLC) from Lau et al. (2023) before defining the refined variants that are new to this paper. The LLC at a neural network parameter $w^*$, denoted $\lambda(w^*)$, is a positive scalar measuring the degeneracy of the geometry of the population loss $\ell$ near $w^*$. The geometry is more degenerate (lower LLC) if there are more ways in which $w$ can be varied near $w^*$ such that $\ell(w)$ remains equal to $\ell(w^*)$.

### 3.1 LLC IN PRACTICE

In the setting of Section 2, where we have a compact parameter space $W$, a model with parameter $w$ of the conditional distribution of outputs $y$ given inputs $x$ (in our case a transformer neural network with weights $w$ parametrizes predictions of next-tokens $y$ given contexts $x$), and samples $D_n$ from a true distribution with associated empirical loss $\ell_n$, we define the *estimated local learning coefficient* at a neural network parameter $w^*$ to be

$$\hat{\lambda}(w^*) = n\beta \left[ \mathbb{E}_{w|w^*,\gamma}^{\beta}[\ell_n(w)] - \ell_n(w^*) \right], \tag{2}$$

where $\mathbb{E}_{w|w^*,\gamma}^{\beta}$ is the expectation with respect to the Gibbs posterior (Bissiri et al., 2016)

$$p(w; w^*, \beta, \gamma) \propto \exp\left\{ -n\beta\ell_n(w) - \frac{\gamma}{2}||w - w^*||_2^2 \right\}. \tag{3}$$

The hyperparameters are the sample size $n$, the inverse temperature $\beta$ which controls the contribution of the loss, and the localization strength $\gamma$ which controls proximity to $w^*$. For a full explanation of these hyperparameters the reader is referred to Watanabe (2013); Lau et al. (2023); Furman & Lau (2024); Hoogland et al. (2024). Further, the expectation is approximated by using stochastic-gradient Langevin dynamics (SGLD; Welling & Teh, 2011) which introduces additional hyperparameters such as the step size; see Appendix F.2 for the settings used in this paper.

Intuitively, the quantity in (2) represents the typical deviation in empirical loss $\ell_n(w) - \ell_n(w^*)$ under perturbations away from $w^*$ that are likely according to a local tempered posterior distribution. For more theoretical insight into this intuition see Appendix A.

## 3.2 REFINED LLC

The LLC $\lambda(w^*)$ depends on the parameter space and the true distribution. If we view some directions in parameter space at $w^*$ as fixed and view the model as a function of the remaining directions we obtain the *weight-refined* LLC. If we instead allow the true distribution to vary we obtain the *data-refined* LLC. We now explain both in more detail.

**Weight- and data-refined LLC (wdrLLC).** Let $q'$ be a data distribution, not necessarily the training distribution $q$, and let $\ell'$ and $\ell'_n$ be the corresponding population and empirical loss. Given a product decomposition $W = U \times V$ corresponding to choosing a set of weights $V$ belonging to a particular component of the model (with $U$ denoting the rest of the weights), with associated decomposition of the parameter $w^* = (u^*, v^*)$, we let $B$ be a neighborhood of $v^*$ small enough that $\ell'(u^*, v) \geq \ell'(u^*, v^*)$ for all $v \in B$ and define

$$\mathrm{vol}(\epsilon, w^*, V, q') = \int_{v \in B, |\ell'(u^*, v) - \ell'(u^*, v^*)| < \epsilon} dv. \tag{4}$$

The *weight- and data-refined* LLC is

$$\lambda(w^*; V, q') = -\lim_{\epsilon \to 0^+} \log_2 \left[ \mathrm{vol}(\tfrac{1}{2}\epsilon, w^*, V, q') / \mathrm{vol}(\epsilon, w^*, V, q') \right]. \tag{5}$$

The associated estimator $\hat{\lambda}(w^*; V, q')$ is defined by modifying (2) as follows: the expectation $\mathbb{E}^{\beta}_{w|w^*, \gamma}$ is replaced by the expectation with respect to a Gibbs posterior defined over $V$ by

$$p(v; v^*, q', \beta, \gamma) \propto \exp \left\{ -n\beta\ell'_n(u^*, v) - \frac{\gamma}{2} ||v - v^*||_2^2 \right\}. \tag{6}$$

In practice, the estimator is implemented by projecting the SGLD update steps used to produce approximate posterior samples onto $V$ and computing both SGLD updates and average posterior loss using samples from $q'$. For further background on the theoretical definition in (5) see Appendix A.

When $q' = q$, we suppress $q$ and refer to this as the *weight-refined LLC* (wrLLC) $\hat{\lambda}(w^*; V)$, and when $V = W$, we suppress $V$ and refer to this as the *data-refined LLC* (drLLC) $\hat{\lambda}(w^*; q')$. When both $q' = q$ and $V = W$, we recover the original LLC $\hat{\lambda}(w^*)$.

## 3.3 LIMITATIONS

There are numerous limitations to LLC estimation in its present form, including:

- The justification of the LLC estimator $\hat{\lambda}(w^*)$ presumes that $w^*$ is a local minima of the population loss but there is no clear way to ascertain this in practice, and we typically perform LLC estimates during training where this is unlikely.
- Accurate estimates can be achieved in some cases where we know the true LLC (Furman & Lau, 2024), but in general, the ground truth LLC is unknown. As such, we cannot guarantee the accuracy of estimated LLC values in transformer models, but we do have reason to believe that the *ordinality* is correct, e.g. that if the wrLLC estimates of two attention heads are in a particular order, then this is also true of the underlying true wrLLCs.

We expect SGLD-based LLC estimation to mature as a technique. In the meantime, a series of papers (Lau et al., 2023; Chen et al., 2023; Furman & Lau, 2024; Hoogland et al., 2024) have demonstrated that despite these limitations, the estimated LLC does *in practice* seem to offer a useful signal for studying neural network development. In the appendix, we compare our analysis using rLLCs against Hessian-based methods (Appendix D) and ablation-based methods (Appendix E). Some analysis is given for other seeds (Appendix G).

## 4 EMPIRICAL RESULTS

### 4.1 DIFFERENTIATION VIA WEIGHT-REFINED LLCS

The weight-refined LLC for an attention head is a measure of the amount of information needed to specify a configuration of the weights in the head which achieves a certain relative improvement in

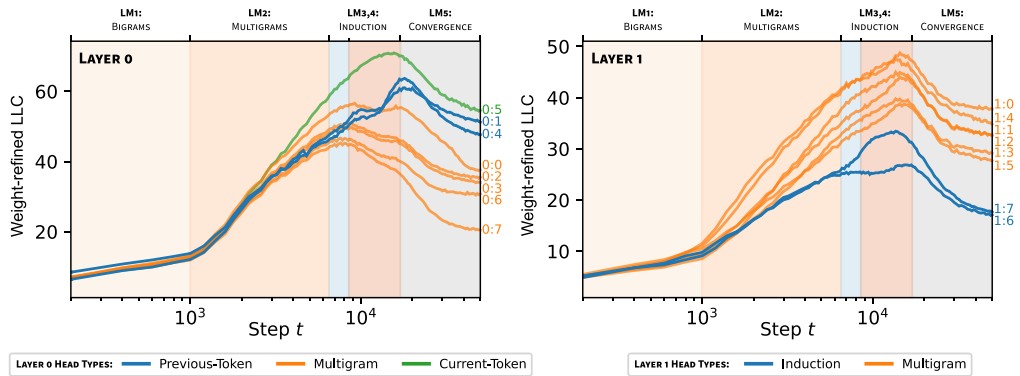

Figure 2: **The weight-refined local learning coefficient (wrLLC) reveals how different types of attention heads *differentiate* during training**. The wrLLC curve for each head is shown colored by its functional type (Appendix B). Remarkably, the partition of the heads by type coincides with the clustering of their wrLLC curves, viewed as time series and clustered by Euclidean $K$-means (Appendix F.5). This suggests that heads which compute differently, develop differently, as revealed by the wrLLC. Throughout this paper, developmental stages LM1–LM5 are colored in the background according to the classification of Hoogland et al. (2024).

the loss (see Section 3.2). Thus we should expect *complexity* to be the principal axis along which the wrLLC differentiates attention heads. However, *a priori* it is not obvious how the complexity as measured by the wrLLC should relate to other properties of the attention heads, such as the classification of heads by their functional behavior or the number of multigrams that they memorize. Our first key contribution, contained in Figure 1 and Figure 2, is to show that there is in fact a very natural relation between the wrLLC and these other axes of differentiation.

As explained in detail in Appendix B, we classify attention heads as previous-token heads (resp. current-token heads) if they strongly and systematically attend to the previous token (resp. current token). Induction heads are identified as in Elhage et al. (2021). All other heads are referred to as *multigram heads*, as ablating them tends to highly impact prediction of multigrams. These categories give the *type* of an attention head.

Figure 2 shows that when the attention heads in both layers are colored by their type, it is immediately visible that heads of the same type tend to cluster together not only in terms of their wrLLC at the end of training but also in terms of the *overall shape of the wrLLC curve across training*. More precisely, curves within a cluster share notable features such as scale, shape, and critical point locations. This visual impression is backed up by the results of various clustering algorithms (Appendix F.5). As a further test that the clusters are semantically meaningful, we examine the subdivisions that occur as the number of clusters is increased in Appendix F.5.3.

In the case of the layer 1 heads, it is further the case that heads with higher wrLLC tend to memorize more multigrams (see Figure 1 and Figure 11), and heads with lower wrLLC tend to be described by simple algorithms like induction or bracket matching (see Figure 1). Thus, the differentiation of attention heads by their wrLLC lines up with an intuitive sense of the description length of the head's computational behavior. For a more in-depth and quantitative treatment, see Appendix B.1.

In summary, the wrLLC reveals a differentiation of the attention heads across training that we can independently verify is semantically meaningful.

## 4.2 SPECIALIZATION VIA DATA-REFINED LLCs

In the previous section we saw that the weight-refined LLC reveals the differentiation of heads into functional types, which are useful for prediction on different kinds of patterns (e.g. induction patterns vs. multigrams). We now demonstrate how further refining the LLC measurements by changing the data distribution (such as to one more heavily featuring certain kinds of patterns) provides additional information on model components specializing to particular patterns in the data.

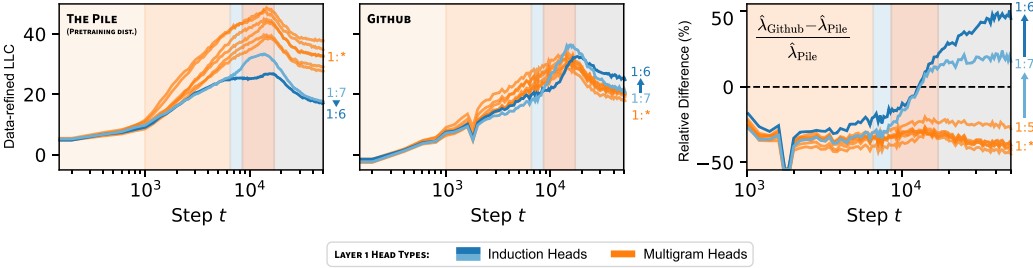

Figure 3: **The data-refined local learning coefficient (drLLC) reveals how attention heads** *specialize* **to different types of data.** The data-refined LLC for GitHub (middle, CodeParrot 2023) indicates that on code samples, perturbations to the weights in the multigram heads in layer 1 have significantly less impact on the loss than perturbations to the induction heads. Informally, the drLLC suggests these heads are differentially more important for predicting code than natural language. This distinction is especially pronounced for 1:6.

If we think of the weight-refined LLC $\lambda(w^*; V)$ as a measure of the information in $V$ about *all* patterns in the pre-training distribution, then the simultaneous weight- and data-refinement $\lambda(w^*; V, q')$ measures the information about the subset of those patterns that occur in a subdistribution $q'$.

For example, when a head $V$ is specialized to multigrams that are uncommon in code we predict that $\lambda(w^*; V, q_{\text{GitHub}}) < \lambda(w^*; V)$ when $q' = q_{\text{GitHub}}$ is a distribution of code (CodeParrot, 2023). In the opposite direction, since induction patterns are frequent in code (e.g. repeated syntax, repeated variable names), we expect that $\lambda(w^*; V, q_{\text{GitHub}}) > \lambda(w^*; V)$ when $V$ is an induction head.

Both of these predictions are borne out in Figure 3. Further, we observe that the wrLLC values of the two induction heads are nearly identical at the end of training, but they are pulled apart by the data-refinement to code, with 1:6 having a significantly higher value. This leads to the prediction that 1:6 is specialized further, *within the set of induction patterns*, to patterns common in code.

We verify this prediction in Figure 4 by examining some examples of contexts on which prediction is most negatively affected by ablating 1:6 and 1:7 and see that, indeed, the former set of examples tend to have a more syntactic or structural flavor relative to the latter, which is consistent with the data-refined LLC for code of 1:6 having a higher value. In Appendix B, we cover more examples validating this behavioral difference, including Figure 10, which shows that changes in rLLC curves accurately reveal that in-context learning develops at different times for each head.

Just as we can decompose a neural network into its architectural components (e.g. attention heads) we can imagine decomposing the pre-training distribution into its structural components (e.g. different data sources). The development process sets up a rich interplay between these two types of components, and the simultaneous weight- and data-refined LLCs quantitatively illustrate which components of a model are being shaped in response to what structure in the data. For example, 1:6 may have been significantly influenced by code.

## 4.3 A NEW MULTIGRAM CIRCUIT

A multigram of length $m$ is a common sequence of tokens $t_1, t_2, \ldots, t_m$ in the data distribution, where the tokens may appear non-contiguously in context (Shen et al., 2006). In this paper, multigrams are typically of length 3 or 4 and often do involve consecutive tokens. It is well-known that transformers can implement the prediction of bigrams ($m = 2$) using the embedding and unembedding layers, and the prediction of skip-trigrams (a subset of $m = 3$ multigrams) using individual attention heads (Elhage et al., 2021). However, the prediction of more complex multigrams may require coordination between attention heads in different layers. In this section, we explain how we used refined LLCs to investigate this coordination and provide evidence for a new circuit involved in multigram prediction.

As noted in Hoogland et al. (2024) and revisited in Figure 13, two-layer attention-only transformers pass through consecutive developmental stages where they behave like zero- and one-layer transformers. It is around stage LM3 that the behavior of the two-layer transformer starts to diverge from that

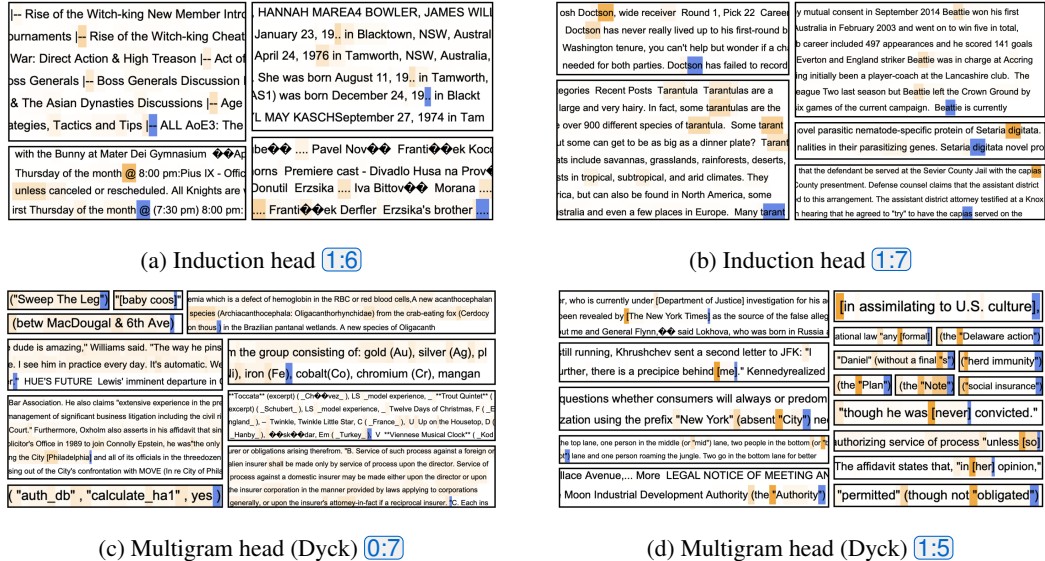

Figure 4: **Induction heads and multigram heads develop subspecializations.** Compared to 1:7, the induction head 1:6 is more involved in predicting induction patterns (Appendix B.2) that feature punctuation and special characters. Multigram heads 0:7 and 1:5 learn skip $n$-grams involved in bracket-matching ("Dyck patterns" Appendix B.3). Blue indicates the token to be predicted. Orange indicates the strength of the attention pattern at the current token. Samples are selected by filtering for tokens where ablating the given head leads to the largest increase in loss.

of a one-layer transformer. Thus, if it exists, coordination between heads for complex multigram prediction is likely to emerge in LM3.

The development of such coordination might require "displacing" learned structure for the prediction of simpler multigrams. To test this hypothesis, we investigated how the information in attention heads about simple multigrams changes over training by using data-refined LLCs with a one-layer transformer as the generating process for the data distribution (see Appendix F.3). These drLLCs are shown in Figure 5 and compared with the wrLLCs for the full pre-training distribution. In line with our expectations, we see that the two sets of LLCs are similar early in training and start to diverge around LM3, which we interpret as a relative decrease across all attention heads of the information about simple multigrams that can be predicted by the one-layer model.

If we examine the cluster of heads with the largest relative decrease, we see some examples (e.g. the previous-token, current-token and induction heads, classified as such according to their behavior at the end of training) for which the decrease is easily explained: they are involved in predicting simple multigrams at $t = 8.5k$ steps but acquire other roles by the end of training (see Figure 9 and Appendix B.6). However it is *a priori* surprising to find the layer 0 multigram heads in this cluster, since ablation experiments indicate that they are primarily involved in multigram prediction throughout training.

This suggests that it is in these layer 0 multigram heads that simpler multigrams are displaced by the development of coordination for more complex multigram prediction. We use mean ablations to test this theory. In Figure 4, we see that the head 1:5 seems to involve Dyck patterns (closing parentheses and brackets, see Appendix B.3). Curiously, ablating 0:7 at the end of training heavily impacts the same kinds of patterns. However, this is not the case earlier in training: at training step $t = 8.5k$, 0:7 is responsible for a different set of multigrams (Appendix B.6). Combined with the observation from Figure 5, this leads to the hypothesis that this layer 0 multigram head may be going through some transition that that causes it to switch to primarily passing information forward to layer 1 multigram heads, including 1:5.

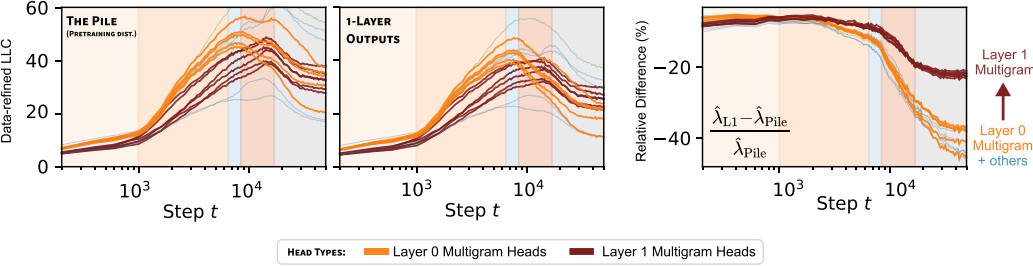

Figure 5: **Using a 1-layer (L1) model as the data distribution for data-refined LLCs helps locate skip-trigram-related structure.** During stage LM3, the drLLC begins decreasing for layer 0 multigram heads while increasing for layer 1 heads (middle). In stage LM5, when layer 1 drLLCs also start decreasing, the decline is significantly less pronounced than in the layer 0 multigram heads and the rest of the heads (right).

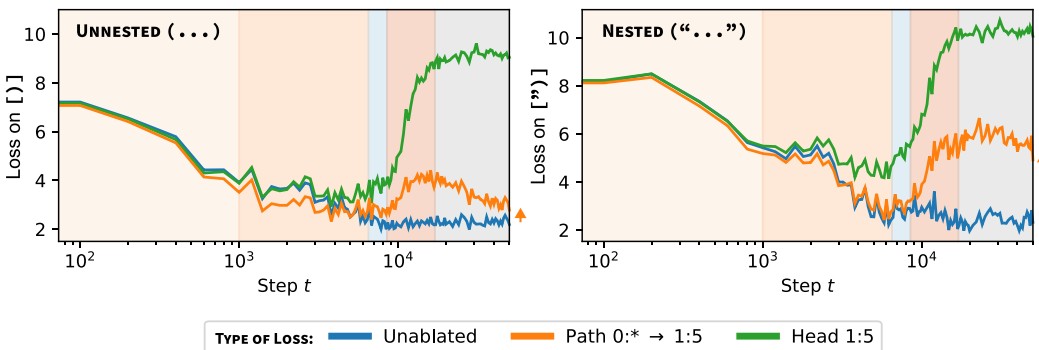

Figure 6: **The formation of the multigram circuit enables nested bracket-matching.** Head $\boxed{1:5}$ is a multigram head that specializes to matching brackets. On a synthetic dataset of sentences with parentheses, mean-ablating this head causes loss to increase sharply (blue line to green line). With nested brackets (right), this may require coordination with the layer 0 multigram heads; mean-ablating the multigram heads and patching their activations just into the input of $\boxed{1:5}$ (Appendix E.2) causes an increase in loss (orange), *precisely when we predict the shift in computational role occurs*. With unnested parentheses, $\boxed{1:5}$ does not need the layer 0 multigram heads; the same procedure leads to a decrease in performance during LM4, but by the end of LM5, the effect is small.

We validate this hypothesis with path patching (Wang et al., 2023) in Figure 6, where we note a change in interaction between layer 0 multigram heads and $\boxed{1:5}$ at the expected time. In Appendix B.6, we present examples of multigrams migrating from layer 0 to layer 1 between $t = 8.5k$ and $t = 50k$.

To further corroborate the hypothesis that layer 0 and layer 1 multigram heads are coordinating, we check their K-composition scores, defined by Elhage et al. (2021). Informally, these measure how much an attention head in a later layer reads from the write subspace of the residual stream of an attention head in an earlier layer (for a more formal definition, see Appendix F.4). In Figure 7, we see that the layer 0 multigram heads all undergo the same turnaround in K-composition scores with the layer 1 multigram heads, including an unusually strong coupling of composition scores from a given layer 0 head to each of the layer 1 multigram heads. We refer to this pattern of coordination between layer 0 and layer 1 multigram heads as the *multigram circuit*.

In summary, these results suggest that layer 0 and layer 1 multigram heads independently memorize simple multigrams until around the end of LM3. The layer 0 heads then begin forgetting some simple multigrams as they transition to a supporting role within the multigram circuit. At the same time, the layer 1 multigram heads specialize to more complex multigrams, such as nested Dyck patterns.

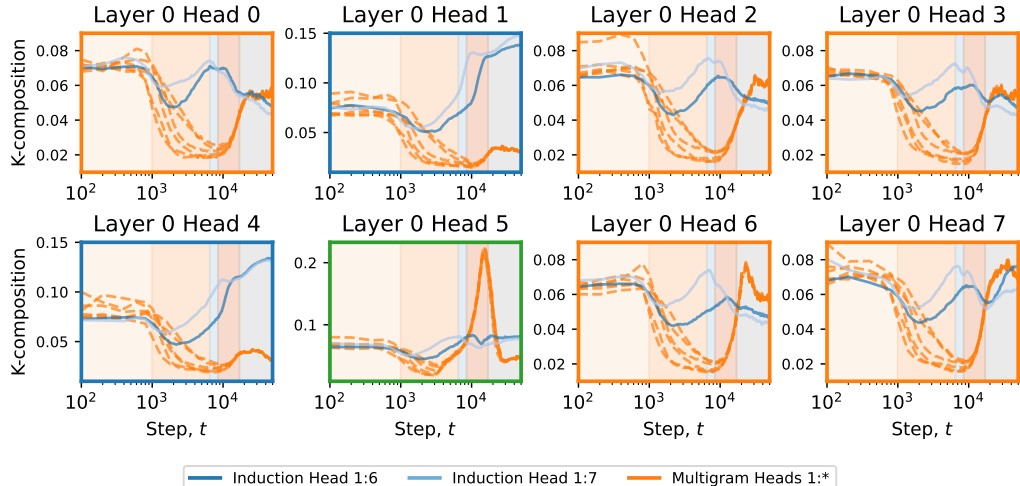

Figure 7: **K-composition scores between layer 0 multigram heads and layer 1 multigram heads increase in tandem during stage LM4**. After a decrease in LM2, K-composition between multigram heads begins increasing during LM4, while composition between multigram and induction heads begins decreasing. Subfigures correspond to heads in layer 0. Individual lines show K-compositions over training between that head and a layer 1 head. Border color indicates head type: orange for multigram heads, blue for previous-token heads, and green indicates for current-token head.

## 5 RELATED WORK

**Data distributional structure.** It is clear that structure in the data distribution plays a significant role in the kinds of structures learned in neural networks and *how* they are learned (Rumelhart et al., 1986; Olshausen & Field, 1996; Rogers & McClelland, 2004). For instance, properties of the data distribution have been linked to the emergence of in-context learning by Chan et al. (2022b), and Belrose et al. (2024) note that networks learn lower-order moments before higher-order ones.

In the experiments and accompanying theory of Rogers & McClelland (2004, p.103, p.169), "waves of differentiation" are triggered by coherent covariation in the data distribution. They claim that the "timing of different waves of variation, and the particular groupings of internal representations that result, are governed by high-order patterns of property covariation in the training environment" (Rogers & McClelland, 2004, p.103). Some of these ideas were given a theoretical basis in Saxe et al. (2019) which contains a mathematical model of Rogers & McClelland (2004).

**Specialization.** Krizhevsky et al. (2012) observed that in the first layer of AlexNet, the two branches specialized to different kinds of features. Voss et al. (2021) later hypothesized that this phenomenon of *branch specialization* is driven ultimately by structure in the data. Some of the earliest work in mechanistic interpretability (Cammarata et al., 2020) demonstrated interesting specialization in convolutional neural networks. A more automated search for local specialization was pursued in Filan et al. (2021); Hod et al. (2022); Casper et al. (2022).

## 6 DISCUSSION

In this paper we have introduced the refined LLCs (rLLCs) as a principled tool for understanding internal structure in neural networks and shown that this tool can be used to study the differentiation and specialization of attention heads over training. This builds on the theoretical foundations of SLT(Watanabe, 2009), the introduction of the ordinary LLC (Lau et al., 2023) and recent results showing that changes in the LLC over training reflect developmental stages Hoogland et al. (2024).

This section puts these contributions into the broader context of the science of deep learning and interpretability. From a structural point of view, the problem of interpretability for neural networks is

to understand internal structure and how it determines the map from inputs to outputs. We take the point of view that this problem cannot be solved in a deep way without first addressing the question: what is the true conception of internal structure in neural networks?

There is a long tradition in mathematics (Langlands, 1970), computer science (Howard et al., 1980) and physics (Maldacena, 1999; Greene & Plesser, 1996) of understanding the nature of a mathematical object or phenomena by putting it in correspondence or duality with other phenomena. It is therefore interesting to note the literature (reviewed in Section 5) arguing that data distributional structure is an important factor in shaping internal structure in neural networks, and that this structure is further linked to structure in the learning process. To this we may add the singular learning theory perspective, which relates geometric structure of the population loss landscape to the structure of the (singular) learning process (Watanabe, 2009; Chen et al., 2023).

The synthesis of these perspectives suggests a novel approach to the problem of understanding the fundamental nature of internal structure in neural networks, which is to place the problem in the broader context of studying the correspondence between four categories of structure:

- **Data distributional structure**: the inherent patterns and regularities in the data which exist independently of any model (Cristianini & Shawe-Taylor, 2004). *In this paper*: induction patterns, Dyck patterns and multigrams (Appendix B.1).
- **Geometric structure**: the analytic and algebraic geometry of the level sets of the population loss (Watanabe, 2009; Amari, 2016). *In this paper*: the learning coefficient (or real log canonical threshold, as it is known in geometry).
- **Learning process structure**: developmental stages, critical periods, and the sequence in which different capabilities or internal structures emerge (Rogers & McClelland, 2004). *In this paper*: the overall developmental stages of (Hoogland et al., 2024) and the staggered development of individual attention heads.
- **Computational structure in the model**: the functional organization within the neural network itself and computational motifs that emerge during training. *In this paper*: attention heads, the induction and multigram circuits.

Here by *structure* we loosely mean the "arrangement of and relation between parts or elements of something complex" (McKean, 2005). Since there can be no structure without differentiation of the whole into parts, the foundation of these correspondences is a relation between the "parts or elements" in each of the four categories. From this perspective, **the contribution of the present paper is to begin establishing such a correspondence for two-layer attention-only transformers** by using refined LLCs to quantitatively track which components of the model are being shaped in response to what structure in the data distribution. More precisely:

- In Section 4.1 we related computational structure (the behavioral type of attention heads) to learning process structure and geometric structure as measured by the LLC, by showing that the behavioral type can be recognized by clustering wrLLC curves (Figure 1 and Figure 2).
- In Section 4.2 we related data distributional structure (the difference between the frequency of certain kinds of induction patterns in code versus natural language) to the differences in geometry between particular induction heads (Figure 3).
- In Section 4.3 we related data distributional structure (nested brackets in Dyck patterns) to a new multigram circuit whose emergence (a structure in the learning process) seems linked to geometric changes in the layer 0 multigram heads (Figure 5).

Finally, we highlight that many of these results depend on a developmental perspective. We could not cluster attention heads by their weight-refined LLCs without seeing their evolution during training, nor could we clearly see the connection between induction patterns and performance on code samples without observing changes in the data-refined LLC curves. Even the ablation and composition score analyses of the multigram circuit depend on comparing results across training.

The techniques pioneered in this paper for understanding internal structure in two-layer attention-only transformers can be applied to models at a larger scale or with different architecture. We refer to this approach, which combines the refined LLCs from singular learning theory with an emphasis on studying networks over the course of development, as developmental interpretability. Using this set of ideas, we hope to open new paths towards a systematic understanding of advanced AI systems.

## REPRODUCIBILITY STATEMENT

Detailed descriptions of our experimental setup, including model architecture, training procedures, and hyperparameters, are provided in Appendix F.

Our LLC estimation procedure is documented in Appendix F.2, which lists the SGLD hyperparameters used for estimating the Local Learning Coefficient and references detailed resources for implementing LLC estimation. Methodological details for implementing the various refinements are provided in Section 3 and Appendix F.3.

Details for the procedure used to classify heads are provided in Appendix B.1. We also provide a link to an anonymized repository that contains extended results of this analysis along with additional figures. More implementation details for ablation-based analyses are provided in Appendix E.2. Implementation details for the comparison with Hessian-based metrics (Appendix D) can be found in the associated subsections.

## CONTRIBUTION STATEMENT

The following is a non-exhaustive list of some particular areas of individual contribution.

- GW led the project, developed the methodology for LLC hyperparameter calibration, identified the multigram circuit, conducted experiments using other interpretability techniques, contributed to the classification of attention heads, and made substantial contributions to manuscript writing.
- JH contributed to engineering, led the classification of attention heads, performed clustering analyses, designed figures, and made substantial contributions to manuscript writing.
- SvW led the engineering and conducted the LLC estimation experiments.
- ZF conducted the experiments comparing against Hessian-based methods, including design of methodology and associated writing and figures.
- DM had the original idea for refined local learning coefficients, played a role in identifying the multigram circuit, and contributed substantially to writing the main text.

## ACKNOWLEDGEMENTS

We thank Matthew Farrugia-Roberts for his feedback on earlier drafts of this manuscript.

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

APPENDIX

**Appendix A** provides theoretical background on the local learning coefficient.

**Appendix B** provides further details on the head classification. We describe the methodology we followed to manually classified each head, and offer more explanation and examples of each head's classification and specializations.

**Appendix C** discusses the significance of critical points, increases, and decreases in the (r)LLC in relation to stagewise development in artificial and biological neural networks. We provide additional results for the full-weights data-refined LLC for a variety of common datasets.

**Appendix D** compares the rLLC to the Hessian trace, Fisher Information Matrix (FIM) trace, and Hessian rank. We show that the LLC consistently outperforms these other techniques.

**Appendix E** compares the rLLC against ablation-based metrics that are common to mechanistic interpretability (zero ablations, mean ablations, and resample ablations). We discuss the strengths and weaknesses of each of these methods in relation to the rLLC.

**Appendix F** provides more experimental details on the architecture, training setup, LLC hyper-parameters, expected cost of scaling the methodology, using models as the generating process for data-refined LLCs, composition scores, and automated clustering analysis.

**Appendix G** examines the consistency of our findings across different random initializations, supporting the generality of our conclusions. This analysis further supports the robustness of our observations across various model components.

**Appendix H** examines consistency across models scales, rerunning the analysis of the induction circuit for Pythia-70m (Biderman et al., 2023).

**Additional figures & data** can be found at the following anonymized repository: anonymous.4open.science/r/paper-rllcs-2024-BE57.

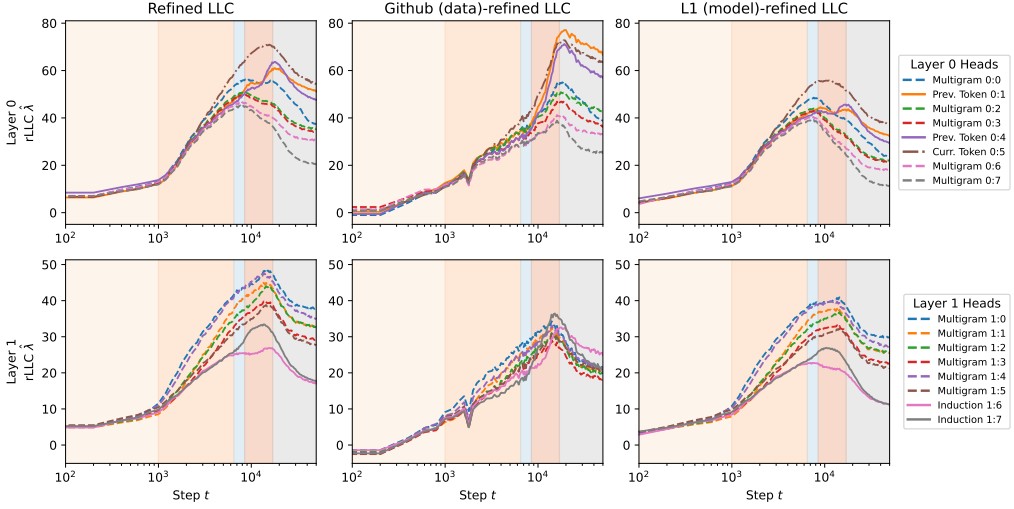

Figure 8: **Detailed reference for per-head rLLCs.** This figure combines data from figures Figure 2, Figure 3, and Figure 5 for detailed cross-referencing.

# A  LLC IN THEORY

What is not necessarily clear from (2) is that this is an estimator for a theoretical quantity $\lambda(w^*)$ which is an invariant of the geometry of the population loss. To explain we recall one form of the definition of the learning coefficient from Watanabe (2009). For a triple $(p, q, \varphi)$ consisting of a parameter space $W$ with model $p(y|x, w)$, truth $q(y|x)$ and prior $\varphi$ on $W$ we consider the volume

$$\text{vol}(\epsilon) = \int_{K(w) < \epsilon} \varphi(w) dw$$

where $K(w) = \int D_{KL}(q(y|x)||p(y|x, w))q(x)dx$. Under some conditions (Watanabe, 2009, Theorem 7.1) the learning coefficient is given by

$$\lambda = - \lim_{\epsilon \to 0} \log_2 \left[ \text{vol}(\tfrac{1}{2}\epsilon)/\text{vol}(\epsilon) \right]. \tag{7}$$

The *local* learning coefficient $\lambda(w^*)$ at a local minima $w^*$ of $K(w)$ is defined by restricting the volume integral to a neighborhood $B$ of $w^*$ where $K(w) \geq K(w^*)$

$$\text{vol}(\epsilon, w^*) = \int_{w \in B, |K(w) - K(w^*)| < t} \varphi(w) dw \tag{8}$$

and then defining (Lau et al., 2023; Furman & Lau, 2024)

$$\lambda(w^*) = - \lim_{\epsilon \to 0} \log_2 \left[ \text{vol}(\tfrac{1}{2}\epsilon, w^*)/\text{vol}(\epsilon, w^*) \right]. \tag{9}$$

This is the asymptotic number of bits necessary to specify a parameter near $w^*$ which is half again closer to the truth. This has some relation to intuitive notions of "flatness" (Hochreiter & Schmidhuber, 1997) or description length (Grünwald, 2007).

In practice we do not have access to the function $K$ since it depends on the true distribution. Nonetheless there are several methods available to estimate these quantities empirically (Watanabe, 2013; 2009), using the negative log-likelihood of a set of samples $D_n$. In practice we substitute the population loss $\ell$ for the KL divergence $K$ and use (2) to approximate the LLC, rather than try to directly approximate (9). Nonetheless this formula offers a valuable information-theoretic interpretation of the LLC that we employ in this paper.

# B  CLASSIFICATION OF ATTENTION HEADS

In this section, we classify attention heads by their behavior. At a high level, the attention heads can be thought of as being a part of one of four groups, illustrated in Table 1. Unless specified otherwise, all description of attention heads refers to behavior at the end of training.

## B.1  METHODOLOGY: TOKENS IN CONTEXT

**Identifying tokens in context.**  To identify which patterns in data each attention head is associated to (independently of rLLCs), we mean-ablate that head (Appendix E), then filter a subset of 100k samples from the training dataset for the 1k most affected "tokens in context" (i.e., a sample index combined with a position index), as measured by the increase in the per-token loss. This results in pairs of a token in context and local attention pattern for the ablated attention head that generated that sample.

**Classifying tokens in context.**  We attempt to classify each (token in context, attention pattern) pair as belonging to one of the following "patterns:"

- **Induction patterns** (Appendix B.2)
- **Dyck patterns** (Appendix B.3)
- **Skip $n$-grams** (Appendix B.4)
- **$n$-grams** (Appendix B.5)

| Head | Classification | Comments | Section |
|---|---|---|---|
| 0:0 | Multigram | $n$-grams esp. prepositional phrases | B.5 |
| 0:1 | Previous-token | Esp. Proper nouns | B.2 |
| 0:2 | Multigram | $n$-grams; Post-parenthetical tokens | B.5 |
| 0:3 | Multigram | $n$-grams | B.5 |
| 0:4 | Previous-token | Esp. dates and multiple spaces | B.2 |
| 0:5 | Current-token | | B.6 |
| 0:6 | Multigram | $n$-grams; Post-quotation tokens | B.5 |
| 0:7 | Multigram (Dyck) | (Nested) bracket-matching; Double spaces / start of sentences | B.3 |
| 1:0 | Multigram | $n$-grams, esp. numbers & prices (thousands place), hyphenated phrases ("first-ever", "year-ago"), "[#] [timespan] ago" | B.5 |
| 1:1 | Multigram | $n$-grams, esp. proper nouns; double spaces, comparison completion (more...th -> an) | B.5 |
| 1:2 | Multigram | $n$-grams, esp. dates & start of sentence | B.5 |
| 1:3 | Multigram (Dyck) | Correlative conjunctions ("neither"..."nor"), Abbreviations ("N.A.S.A."), Superlatives ("most"..."ever"), Questions ("Why"..."?"), "..." -> ' | B.3 |
| 1:4 | Multigram | $n$-grams, esp. news-related (date ranges, "Associated press", "copyright", "spokeswoman,"); phrasal verbs ("prevent...from", "keep...safe"), post-quotation tokens ("... '...' said"). | B.5 |
| 1:5 | Multigram (Dyck) | [...], (...), (..."...") {...}, "...", '...', **...** | B.3 |
| 1:6 | Induction (Code) | [A][B]...[A] -> [B] | B.2 |
| 1:7 | Induction | [A][B]...[A] -> [B] | B.2 |

Table 1: Attention Head Taxonomy

This classification is automated and serial: if a token in context cannot be classified as an induction pattern, we subsequently check if it can be described as a Dyck pattern, then a skip $n$-gram, then an $n$-gram. The criterion for inclusion varies for each pattern (and is described in the associated subsection) but typically consists of checking whether the token receiving maximum attention ("max-attention token"), current token, and next token match a particular template. If a pattern matches, we say that pattern "explains" the token in context.

**Classifying heads by tokens in context.** We classify each head by whichever pattern explains the most of its tokens in context, in a relative sense, rather than an absolute sense (so 30% Dyck patterns, 10% skip $n$-grams, 20% $n$-grams, 40% unexplained is enough to classify a head as a Dyck head). This has obvious limitations: there could be another missing pattern that explains the remaining unexplained patterns. We therefore supplement this classification with manual inspection of both explained and unexplained tokens in context (see Figure 9).

We begin by identifying previous-token and induction heads (0:1, 0:4, 1:6, and 1:7), which we then subdivide based on their previous-token and induction scores. The remaining heads are classified as multigram heads that memorize Dyck patterns, skip $n$-grams, and contiguous $n$-grams. Notably, by the end of training, all heads have almost all of their tokens in context explained by these classifications (in some cases only after additional manual inspection). The results of this classification process are displayed in Table 1. For full analyses and datasets of tokens in context, we refer readers to our anonymized repository: https://anonymous.4open.science/r/paper-rllcs-2024-BE57.

**Counting multigrams.** To count the "# Multigrams" in Figure 1 and Figure 11, we add up the unique number of Dyck patterns, skip $n$-grams, and $n$-grams associated to each head. These unique counts are defined in their respective sections.

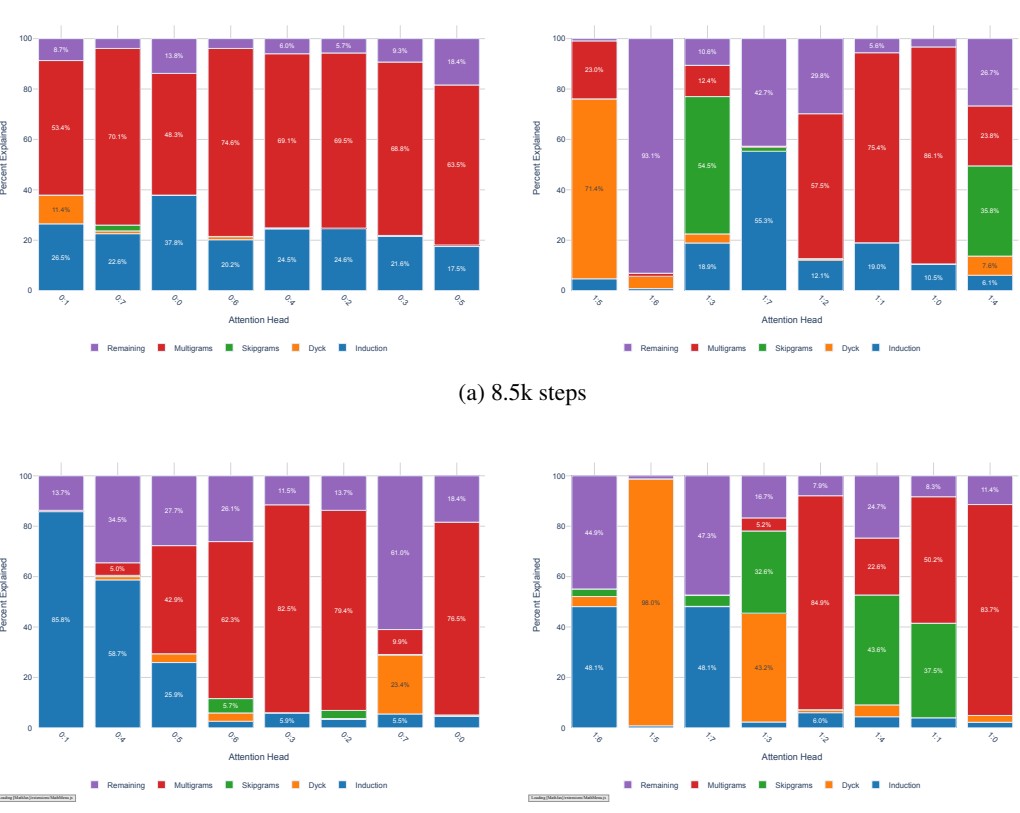

(a) 8.5k steps

(b) 50k steps (Final Checkpoint)

Figure 9: **Token-in-context attributions for each head at 8.5k steps (top row) and at 50k steps (bottom row)**. Each column corresponds to a single head: the bars (left axis) show the percentage of tokens in context that are explained by either induction patterns, Dyck patterns, skip $n$-grams, or $n$-grams (as described in Appendix B.1). The heads are ordered by the number of unique multigrams for each head.

## B.2 INDUCTION PATTERNS

An induction pattern is a sequence of tokens that repeats within a given context, where the model learns to predict the continuation of the repeated sequence. The simplest form of an induction pattern is an in-context bigram, [A][B] ... [A] → [B], where [A] and [B] are arbitrary placeholder tokens. In this pattern, after seeing the sequence [A][B] once, the model learns to predict [B] when it encounters [A] again later in the context. In our analysis, we extend the definition of induction patterns to include arbitrary in-context $n$-grams of the form ([1]...[N]) ... ([1]...[N-1]) → [N].

### B.2.1 CLASSIFICATION

Olsson et al. (2022) showed that two-layer attention-only transformers can develop an *induction circuit* which completes such patterns, predicting the second [N] from the second [N-1]. This circuit involves two components:

1. A *previous-token head* in layer 0, which attends to the second [N-1].

2. An *induction head* in layer 1, which attends to the first [N].

We classify a token in context as part of an induction pattern if it fits this form: if (1) the next token is the second [B]) and (2) the max-attention token is the second [A] (previous-token head) or the first [B] (induction head).

### B.2.2 RESULTS

Hoogland et al. (2024) identified heads `0:1` and `0:4` as previous-token heads and `1:6` and `1:7` as induction heads, using the previous-token score and induction score introduced in Olsson et al. (2022). These heads can also be identified as such by their tokens in context, as illustrated by the selection in Figure 4 and the automated classification in Figure 9.

**Bonus: The rLLC reveals staggered development of induction heads.** The Github drLLC reveals not only that the heads specialize to different data, but also that they form at different times: Figure 10 shows that the inflection point and peak in the rLLC is several thousand steps later for head `1:6` than head `1:7`. To validate this, we consider the in-context learning (ICL) score from Olsson et al. (2022), which takes the average loss at the 500th token minus the average loss at the 50th token.

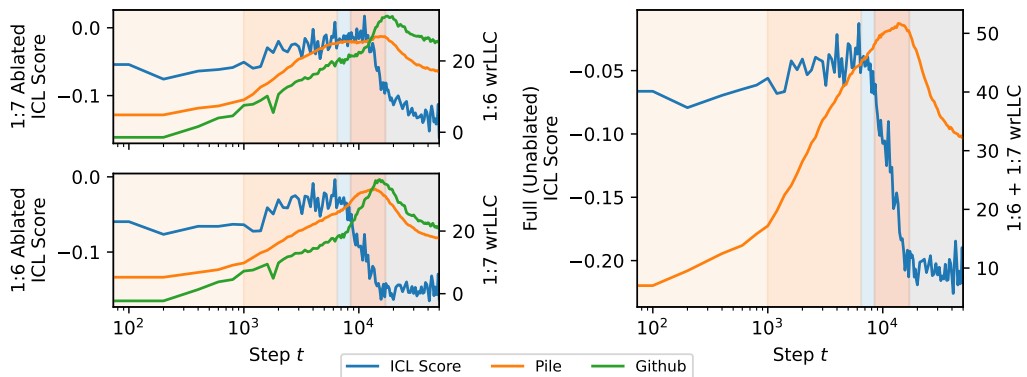

Figure 10: **Induction head development correlates with critical points in the Github drLLC.** The two induction heads do not form at the same time; `1:7` precedes `1:6` by several thousand steps, as is visible in the inflection points of both the Github drLLC for these heads and the per-head ICL score (obtained by mean-ablating the opposite induction head, left column). When restricting to both heads' weights at the same time, the critical point in the Github drLLC coincides with the end of the ICL score drop (right column). This suggests (critical points in) rLLCs can be used to study the development of model components.

We plot the ICL score upon mean ablating either of the two heads: we see that the drop in the ICL score upon mean ablating head `1:7` coincides with the peak in the `1:6` Github drLLC, and vice-versa. Likewise, the peak in the rLLC restricted to both heads' weights at the same time coincides with the standard ICL score.

## B.3   DYCK PATTERNS

In predicting the next token in natural language, some of the most explicit occurrences of hierarchy come in the form of nested brackets and punctuation. This is formalized in the context-free grammars Dyck-$k$ ("deek"), which involve sequences with matching brackets of $k$ kinds (Chomsky & Schützenberger, 1959).

We take a wide definition of bracket, including:

- **Delimiters**:
    - **Traditional brackets/braces/parentheses**: `(...)`, `[...]`, `{...}`, `<...>`,
    - **Quotation marks**: `"..."`, `'...'`, `"..."`, as well as mistokenized `��...��`,
    - **Markdown formatting symbols**: `_..._`, `**...**`,
- **Split constructions**:
    - **Correlative conjunctions**: "both...and", "not only...but also", "neither...nor", "either...or",
    - **Correlative Comparatives**: "more...th", "less...th", "better...th",
    - **Correlative Superlatives**: "most...ever", "least...ever",
    - **Questions**: "Why...?", "What...?", "How...?".

There are similar constructions that we did not analyze but that would make natural candidates for follow-up analysis, such as additional correlative comparatives ("as ... as") conditional statements ("if ... then"), cleft sentences ("it is ... who"), comparative correlatives ("the [more you practice], the [better you get]"), result clauses ("[it was] so [cold] that [the lake froze]"),

The number of unique Dyck patterns for each head is the size of the set of unique (opening token, closing token) pairs for each token in context that is classified as a Dyck pattern.

### B.3.1   CLASSIFICATION

We classify a token in context as part of a Dyck pattern if (1) the next token contains a closing bracket corresponding to an earlier opening bracket, and (2) the subsequence spanned between those two brackets has valid nesting.

Often, we find that the max-attention token is the corresponding opening bracket, especially for layer 1 Dyck heads. However, we do not require this to be true to classify a token in context as a Dyck pattern.

### B.3.2   RESULTS

Figure 9 shows that automated classification identifies three heads as Dyck heads: `1:5` (98% explained), `1:3` (43.2%), and `0:7` (23.4%). Manual inspection of their tokens in context confirms these diagnoses and reveals further subspecializations.

`1:5` **Delimiter-matching head.**   Head `1:5` is specialized to traditional brackets/braces/parentheses, quotation marks, and other symbolic delimiters.

`1:3` **Split-construction-matching head.**   Head `1:3` is specialized to the natural language Dyck patterns, and one variant of quotation marks.

`0:7` **Dyck-support head.**   Head `0:7` is specialized to similar tokens in context as `1:5`. However, only about a quarter of `0:7`'s tokens in context are recognized by our automatic procedure as Dyck patterns. Manual inspection reveals that many remaining unexplained tokens in context are either

misclassified as non-Dyck (e.g., because we identify whether quotation marks are opening or closing by checking whether the previous/subsequent character is not a letter, which is sometimes too restrictive) or Dyck-like (e.g., all-caps text surrounded by multiple spaces, where the spaces serving as delimiters). Most of this head's other remaining tokens in context seem to be involved in skip $n$-grams (analyzed in the next subsection).

One additional way in which these heads are different is their max-attention tokens: the layer 1 Dyck heads both primarily attend to the opening token corresponding to the closing token being predicted. Head `0:7` casts a more diffuse attention pattern.

### B.3.3 DISCUSSION

The ability to correctly close brackets underlies all nonregular context-free languages, in the formal sense that by the Chomsky-Schützenberger theorem, any context-free language arises from a variant of Dyck-2 through intersection with a regular language and homomorphisms (Chomsky & Schützenberger, 1959). For this reason the ability of transformers to recognise Dyck languages has been studied at some length (Hahn, 2020; Bhattamishra et al., 2020; Ebrahimi et al., 2020). In Weiss et al. (2021) an algorithm in RASP is given which compiles to a transformer which recognises Dyck-$k$ languages for any $k$. We did not examine how this relates to the heads investigated here. It is unclear whether transformers actually learn similar algorithms to these in practice (Wen et al., 2023).

As far as we know this paper is the first time that a circuit recognizing a nested Dyck language has been found "in the wild", that is, in a transformer trained on natural language. This seems interesting in connection with the ability of transformers to learn the hierarchical and recursive structure in natural language. It is however well-known that, in general, syntactic structure in natural language is represented within transformers (Hewitt & Manning, 2019).

### B.4 SKIP $n$-GRAMS

The simplest form of a skip $n$-gram is a skip trigram of the form `[A]...[B] -> [C]`, where the distance between tokens `[A]` and `[B]` is variable. We consider more general skip $n$-grams of the form `[1]...([2]...[N-1]) -> [N]`, as well as skip $n$-grams involving multiple steps `[1]...[2]...[N-1] -> [N]`.

The Dyck patterns considered in the previous setting are a special case of skip $n$-grams.

### B.4.1 CLASSIFICATION

We match tokens in context against a preset list of skip $n$-grams, including:

- **Post-quotations**: e.g., `"..." said`, "..." `... says`, �� `...` �� `wrote`, and �� `...` �� `of` (for use with mistokenized scare quotes rather than direct quotations),
- **Post-parentheticals**: e.g., `(@...)` `October`,
- **Post-correlative comparative**: Finishing a comparative of the form "more...th" or predicting what comes after "than."
- **Abbreviations (Acronyms/Initialisms)**: "N.A.S.A.", "N.C.", "D.C.", etc.
- **Phrasal verbs / verb-particle constructions (with object insertion)** like "prevent ... from", "keep ... safe", "let ... go". These were compiled by manually inspecting tokens in context.

The unique number of skip $n$-grams for each head is the size of the set of unique (`[1]`, `[2]`, `...`, `[N]`) tuples for each token in context that is classified as a skip $n$-gram. There are likely quite a few other skip $n$-grams that our search neglected.

Though phrasal verbs fall under the category of (natural language) "split constructions", we choose to analyze these separately from the split constructions listed as Dyck patterns for two reasons. First, Dyck patterns typically have *obligatory* closing elements (like matching brackets or question marks), the particle in phrasal verbs is often optional or can be omitted without rendering the sentence ungrammatical (though correlative comparatives and superlatives are sometimes an exception to this

rule). Second, the Dyck patterns are primarily syntactic while phrasal verbs are more lexical in nature, meaning their behavior and interpretation are more closely tied to specific lexical items and idiomatic meanings rather than general syntactic rules (Dehé, 2002).

### B.4.2 RESULTS

One of the remaining heads appear to be *primarily* involved in skip $n$-grams: `1:4` (43.6%). Heads `1:1` (37.5%) and `1:3` (32.6%) are close to being classified as skip-$n$-gram heads but are more involved in $n$-grams and Dyck patterns, respectively. A few other heads are marginally involved: `0:2` (3.2%), `0:6` (5.7%).

`0:2`, `0:6`, `1:1`, `1:4` **Post-Dyck patterns.** Heads `0:2`, `0:6`, and `1:4` are not themselves Dyck heads. Instead, they are involved in predicting tokens that follow closing parentheticals and two variants of closing quotation marks, respectively. Head `1:1` is responsible for finishing correlative comparisons such as "more...th" -> "an". This is not itself a pure Dyck pattern, since the preceding token has already started to close the opening bracket. Head `1:4` specializes in post quotation-marks, such as "... '...' says ..."

`1:3` **Abbreviation head.** In addition to its role in predicting split-construction Dyck patterns, head `0:3` is also specialized to predicting skip $n$-grams of periods in acronyms and initialisms.

`1:4` **Verb-particle phrases & other set phrases.** Head `1:4` seems to specialize in verb-particle phrases and other set phrases, including examples such as *prevent ... from*, *keep ... safe*, *let ... die*, *let ... go*, *meet ... require(ments)*, *remove ... from*, *accused ... of*, *asked ... whether*, and *turn ... into*. Additionally, this head appears to handle other types of set phrases or common patterns, such as *://...* followed by */* (for URLs), *email* followed by @, and *at* followed by @. This head is also involved in recognizing common verb-object pairs such as *solve ... problems*, and *violated ... laws*.

### B.5 $n$-GRAMS

An $n$-gram is a commonly occurring contiguous sequence of $n$ tokens, e.g., a bigram is `[A] ->
[B]`, a trigram is `[A][B] -> [C]`, and so on.

### B.5.1 CLASSIFICATION

For all remaining tokens in context, we classify the sample as an $n$-gram if the subsequence starting at the token the receives maximum attention up to and including the next token occurs in at least two different tokens in context. We enforce no restrictions on the attention threshold or the length of the $n$-gram.

### B.5.2 RESULTS

All multigram heads memorize some number of $n$-grams, but there is clearly an ordering, where Dyck heads memorize the fewest, the skip $n$-gram-specialized heads slightly more, and the remainder (`0:0`, `0:2`, `0:3`, `0:6`, `1:0`, `1:4`) are almost entirely focused on $n$-grams.

`1:1`, `1:2` **Start-of-sentence heads.** Both `1:1` and `1:2` are responsible for predicting multiple contiguous spaces/newlines and tokens that follow these spaces, in contexts where periods are followed by a double space.

`0:0`, `1:4` **Prepositional phrases.** In addition to learning miscellaneous $n$-grams, head `0:0` specializes to prepositional phrases. Many of its tokens in context are prepositions ("of", "by", "with", "to", "from", "with") that appear in set phrases, like "Chamber of Commerce", "plenty of room", "by email", "went to college", "prevented from doing." `1:4` is also involved in some of these. "at", "on a <blank> basis", "as time goes",

`1:0` **Numbers, hyphens, periods of time.** Head `1:0` specializes to predicting numbers and prices, especially following the thousands place, hyphenated phrases such as "first-ever" and "year-ago",

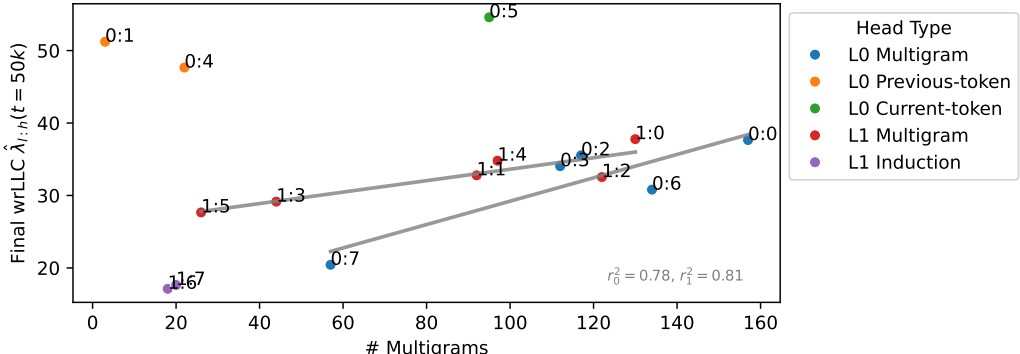

Figure 11: **The # of $n$-grams memorized by each multigram head correlates with the final weight-refined LLC.** By counting the number of unique $n$-grams associated to each head after the head-attribution procedure outlined in Appendix B.1, we find a correlation with the wrLLC.

as well as a period of time ending in "ago" ("years ago", "month ago", "day ago"). It seems to be involved more generally in predicting $n$-grams that are common in a historical or legal context, e.g., "defamation", "disengagement", "argues".

1:4 **News head.** Head 1:4 is especially relevant for predicting $n$-grams that show up in news articles. This includes words like "Associated Press", "spokeswoman" as well as date ranges and post-quotation tokens. It's also involved in several shorter skipgrams involving citations ("- <Name> (@<citekey>) <date>"), attributions ("copyright <Source> image").

### B.6 OTHER OBSERVATIONS

0:5 **Current-token head.** We classify the current-token head 0:5 separately from the above tokens-in-context analysis. This head has two main distinguishing features: it attends almost entirely to the current token, and its composition with the layer 1 multigram heads increases dramatically towards the end of LM4 before decreasing equally dramatically early in LM5.

We do not yet fully understand this head's role. From looking at its tokens in context, this head appears to be involved in predicting several $n$-grams, and potentially also in predicting Dyck patterns (though its attention is concentrated on the current-token and not the opening bracket/quotation mark/etc.).

0:0 **Space head.** Head 0:0 appears to initially become a "space head." At 8.5k steps, its attention pattern tends to distribute itself across all the single-space tokens over the preceding context. This seems to largely revert by the end of training, where it becomes a standard multigram head. This behavior may be linked to why the 0:0-rLLC diverges from the other multigram heads during stages LM1-LM3.

**Migration of the `ago` $n$-gram.** Early in training, at around 8.5k steps, head 0:0 seems to be responsible for predicting time spans ending in "ago," e.g. " years ago" and "one year ago." By the end of training, head 1:0 takes over this role, even though it did not appear to have any involvement at the earlier timestep. Head 0:0 retains only the role of predicting `ago` when it completes a word (as in "Santiago" or "archipelago").

**Migration away from multigram heads.** Heads 0:1, 0:4, 0:7 ultimately become the two previous-token heads and (layer 0) Dyck head respectively. However, these heads appear to start out as simple $n$-gram heads. For example, at 8.5k steps 0:7 (Figure 9) initially learns trigrams like "Fitzgerald", "transient", "transactions", and "transformation" (by running the same procedure to detect tokens in context). At the end of training, these trigrams do not feature among any of 0:7's maximally affected tokens in context. Head 0:7 is also initially involved in predicting the completion of comparative skip-trigrams ("more...th" -> "an"), a role that gets overtaken by 1:1.

## C INTERPRETING CHANGES IN THE (R)LLC

### C.1 INTERPRETING CRITICAL POINTS IN THE (DATA-REFINED) LLC

Hoogland et al. (2024) studied the (non-refined) LLC as a tool for analyzing stagewise development in neural networks. The authors show that critical points in the LLC correspond to boundaries between distinct developmental stages (see Figure 13).

In Figure 12, we show that the full-weights data-refined LLC respects the stage boundaries discovered with the non-refined LLC, for a variety of datasets:

1. **Common training & evaluation corpora**: The Pile (Gao et al., 2020; Xie et al., 2023), TinyStories (Eldan & Li, 2023), and Wikitext (Merity et al., 2016).

2. **Scientific domains** for which we create datasets by filtering Arxiv abstracts by category (Massive Text Embedding Benchmark, 2024): math (`math.*`), physics (`astro-ph.*`, `cond-mat.*`, `gr-qc`, `hep-*`, `math-ph`, `nlin.`, `nucl-*`, `physics.*`, `quant-ph`), computer science (`cs.*`), and economics (`econ.*`).

3. **Human languages** for which datasets sampled from the CC-100 (restricted to languages with a Latin alphabet because the 5k vocabulary size has poor support for other languages). (Conneau et al., 2020; Wenzek et al., 2020)

4. **Programming languages** for which datasets are subsampled from Github (CodeParrot, 2023).

With the exception of the programming languages datasets (discussed in Section 4.2), the drLLCs resemble vertically shifted copies of the original LLC (evaluated on a subset of the Pile). In the scientific domains, the decrease during LM3 is especially pronounced and extends further into LM4 than in the original trajectory. We are not surprised to see that $\lambda_{\text{Math}} > \lambda_{\text{CS}} > \lambda_{\text{Physics}} > \lambda_{\text{Econ}}$.

There are some slight changes in the locations of the boundaries: the decrease in (r)LLC during LM3 flattens out for many of the datasets (TinyStories, natural & programming languages), and it shifts slightly earlier (5.5k-8k steps rather than 6.5k-8.5k steps) for the other datasets. We believe that these changes are more likely to be due to changes in the hyperparameters used for LLC estimation (a slight increase in $n\beta$ from 23 to 30) than related to these particular datasets, see Appendix F.2.

### C.2 INTERPRETING INCREASES IN THE (MODEL-)REFINED LLC

Many works have observed that the training process of small neural networks can have a step-like appearance, with the loss decreasing between plateaus at the same time as a complexity measure

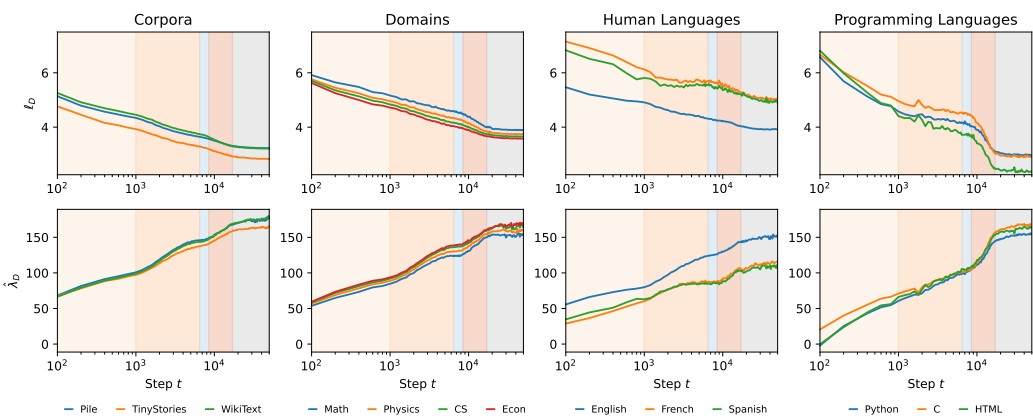

Figure 12: **Data-refined LLC retain coarse developmental stages**. Except for the Github drLLC, the data-refined LLC on its own (without additional weight refinement) yields curves that are close to the original unrefined LLC.

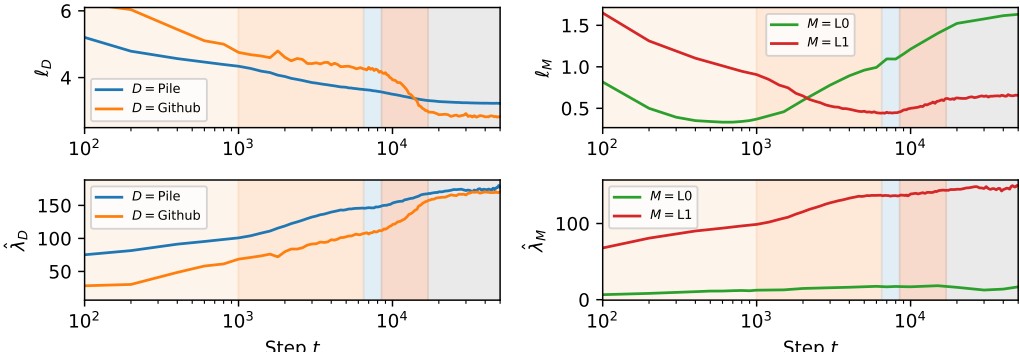

Figure 13: **Two-layer attention-only transformers undergo stagewise development**, learning (LM1) bigrams, (LM2) multigrams, and (LM3 & LM4) the induction circuit, before (LM5) converging (Hoogland et al., 2024). This development may be "hidden" from the training loss (top right, blue line), but discovered by looking for plateaus in the (data-refined) LLC (bottom left, blue line). Alternatively, certain datasets highlight particular developmental stages, e.g., Github (left, orange, CodeParrot 2023) accentuates the development of induction. Likewise, evaluating the model against the outputs of a reference zero-layer (right, green) or single-layer (right, red) model highlights the bigram and multigram stages, respectively. Shown in the bottom right are the data-refined LLCs using zero and one-layer trained transformers as the data generating process (Appendix F.3).

(e.g. rank) increases between plateaus (Arora et al., 2019; Li et al., 2020; Eftekhari, 2020; Advani et al., 2020; Saxe et al., 2019). Indeed, by viewing training as starting at a point of low complexity, this monotonic increase in complexity in simple systems has been put forward as an explanation for the generalization performance of neural networks (Gissin et al., 2019). Similar explanations for generalization performance of infants have appeared in child psychology (Quinn & Johnson, 1997).

In Chen et al. (2023); Furman & Lau (2024) this phenomenon was put into the context of the singular learning process. In this vein, Hoogland et al. (2024) and Figure 12 show that, in simple language models, developmental stages appear to typically (though not always) involve increases in model complexity (as measured by the LLC), with critical points corresponding to some structure or behavior finishing development.

## C.3 INTERPRETING DECREASES IN THE REFINED LLC

The picture of development of neural networks (biological or artificial) in which complexity monotonically increases is incorrect, and thus cannot be a complete explanation for generalization performance. In neuroscience it conflicts with the phenomena of critical periods and synaptic pruning (Kandel et al., 2013; Viviani & Spitzer, 2003), and in neural networks the experiments in Hoogland et al. (2024) show that it is common for developmental stages to involve *decreasing* complexity (as measured by the LLC and mechanistic analysis). This is entirely consistent with the singular learning process.

**Memorization and forgetting.** Intuitively, decreases in the LLC represent an increase in the rigidity of the network's computation (at constant loss) with a corresponding decrease in the LLC, which we can interpret as either increasing geometric degeneracy or decreasing model complexity.

Some of the dyads used to describe similar phenomena in the literature include:

- Memorization and forgetting / reorganization (Achille et al., 2019) in vision networks.
- Memorization / compression (Chen et al., 2024, Appendix G) in language model training.
- Memorization and circuit-formation / cleanup (Nanda et al., 2023) in the setting of grokking (Power et al., 2022).

According to the natural information theoretic interpretation of the LLC (Watanabe 2009, §7.1, Furman & Lau 2024, see also Appendix A) when this quantity is increasing more bits are required

to specify a neighbourhood of a low loss parameter: the amount of information in the weights is increasing. This information could include "memorization" of the inputs. The reverse is true when the LLC decreases, so this is consistent with "forgetting".

However we prefer to avoid these terms since their precise meaning is unclear. A more principled perspective follows from noticing that quantities like the LLC provide upper bounds on both the mutual information between inputs and activations within the network, and the total variation of those activations (Achille & Soatto, 2018, Proposition 5.2). It is consistent with these results that decreasing LLC could be related to a process whereby the representations "discard" extraneous information about the input, and become disentangled; it would be interesting to extend the results of (Achille & Soatto, 2018) using singular learning theory.

**Critical periods.**  In the development of biological neural networks there are periods where temporary "sensory deficits" (interpreted as a shift in data distribution) can lead to permanent skill impairment; these are called *critical periods* (Kandel et al., 2013). It was argued in Achille et al. (2019) that this phenomena is linked to the existence of a "memorization phase" in which the information in the weights increases followed by a "reorganization phase" or "forgetting phase" in which the amount of information is reduced without negatively affecting performance; interventions in the learning process during the memorization phase disproportionately affect the final performance on the test set (making these analogous to the critical periods studied in neuroscience).

In Achille et al. (2019) these two phases were related to *increasing* and *decreasing* trace of the Fisher Information Matrix (FIM), respectively. In our small attention-only language model attention heads undergo periods of rapid specialization followed by periods of refinement, linked respectively to increasing and decreasing weight rLLC; this closely mirrors the concept of critical periods in developmental neuroscience explored in Achille et al. (2019); Kleinman et al. (2023) for vision networks.

In Appendix D.1, we show increases and decreases in the rLLC are correlated with changes in the Hessian trace (a quantity closely related to the FIM trace) during stage LM1, LM2, and LM5, but that they are less reliably correlated for the intermediate stages. In Appendix D.2, we show similar behavior in the FIM trace though this is noisier than the Hessian trace due to methodological limitations.

In Chen et al. (2024) it was observed that an increase and decrease in complexity (measured by e.g. the intrinsic dimension) coincided with an apparent phase transition in the model; this was related to the model of the training process explained in Shwartz-Ziv & Tishby (2017).

**Pruning in neuroscience.**  In developmental neuroscience it is understood that after an initial phase of proliferation during childhood, synapses (the points where signals are exchanged between neurons) are then *pruned*, in a process that accelerates during adolescence and stabilises in adulthood (Johnson & de Haan, 2015, §4.3). The apparent plasticity of the young brain is often attributed to the initial overproduction of synapses and so synaptic pruning is closely related to critical periods.

There are several aspects of pruning that are particularly relevant to the present paper:

- Synapses are kept or pruned based on activity (Johnson & de Haan 2015, §4.3, for a survey see Sakai 2020). This data dependence of pruning has led some to claim that "to learn is to eliminate". Functional structure in the brain is constructed through *growth* and *elimination*, with both being required.

- Some have argued that the differentiation of the brain into separate processing streams and specialized regions is related to synaptic pruning (Johnson & de Haan, 2015, §12.4). A famous example is the emergence of ocular dominance columns (and thus the separation of input from the eyes to facilitate binocular vision) and more general separation of modalities.

- Different regions of the brain have different timings for the reduction of synaptic density. For example the primary sensory areas of cortex show faster growth and decay curves than the prefrontal cortex (Johnson & de Haan, 2015, §4.3).

- It has been suggested that pruning is the biological basis of online Bayesian model selection (Kiebel & Friston, 2011). This is particularly interesting in connection with Watanabe's

discovery (Watanabe 2009, §7.6, see also Chen et al. 2023) that in singular models, internal model selection happens *automatically* as a consequence of free energy minimization.

# D  COMPARISON AGAINST HESSIAN-BASED ANALYSIS

In addition to the LLC, we explore several other geometric measures based on the Hessian of the loss, and evaluate their ability to capture structural development and differentiation over training time. The Hessian, which represents the second-order derivatives of the loss function with respect to model parameters, has been widely used in machine learning to analyze model complexity, optimization dynamics, and generalization properties (LeCun et al., 1989; Dinh et al., 2017).

It is worth noting that the LLC has stronger theoretical foundations as a model complexity measure compared to Hessian-based methods. Unlike Hessian-based methods, the LLC captures higher-order information about the loss landscape, and is not susceptible to high-frequency noise in the empirical loss.

In this section, we investigate three Hessian-based measures:

- Hessian trace (Appendix D.1)
- Fisher information matrix (FIM) trace (Appendix D.2)
- Hessian rank (Appendix D.3)

The Hessian trace shows the greatest empirical success, identifying similar developmental stages as the LLC, albeit with some differences in interpretation. The FIM trace largely agrees with the Hessian trace but introduces additional noise due to estimation methodology. The Hessian rank, while theoretically appealing, proves challenging to estimate reliably and does not clearly reflect the structural development observed through other methods.

Overall, these Hessian-based measures offer complementary perspectives to the LLC analysis, both independently supporting the findings of the LLC, while also highlighting the advantages and disadvantages of the LLC in comparison. The following subsections detail our methodology, results, and discussions for each of these measures.

## D.1  HESSIAN TRACE

We measure the trace of the Hessian of the loss function over the course of training. We measure this for the entire model on the original training distribution, as well as for subsets of the parameters and for different data distributions, similar to what was done for the LLC in Section 4.

The Hessian trace results (Figure 14) show both similarities and differences compared to rLLC findings. It detects some stage transitions and, when weight-refined, differentiates attention head clusters. While the Hessian trace provides valuable insights into model development, its interpretation can be less straightforward than rLLCs in some cases.

### D.1.1  METHODOLOGY

For large models, the Hessian trace can be too expensive to compute explicitly; we use the Hutchinson trace estimator (Hutchinson, 1989; Avron & Toledo, 2011) to efficiently estimate the Hessian trace using Hessian-vector products. This has computational cost comparable to that of the LLC (a constant multiple higher than that of a single SGD step).

The Hutchinson trace estimator relies on the fact that

$$\text{Tr}(H) = \mathbb{E}[v^T H v] \tag{10}$$

for any matrix $H$ and $v$ with entries drawn IID from the Rademacher distribution (Avron & Toledo, 2011). When $H$ is the Hessian matrix, we may efficiently compute the product $Hv$ using standard automatic differentiation techniques, even in cases where explicit computation of $H$ itself is prohibitively expensive. The two hyperparameters of this method are the number of dataset samples to use in calculating the Hessian, and the number of samples used to compute the expectation in Eq 10.

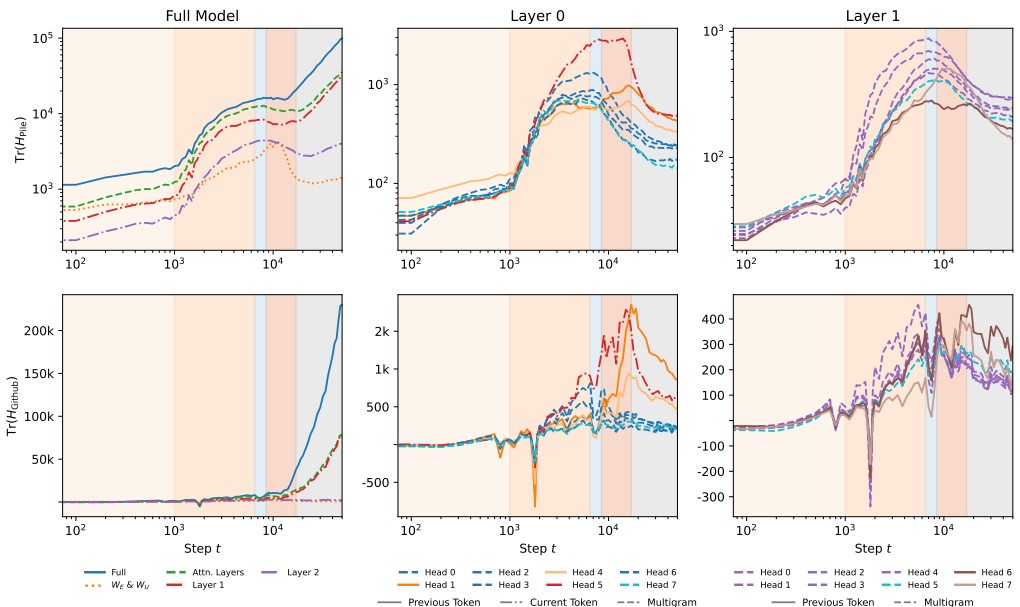

Figure 14: **Estimated Hessian traces** for Pile-13m (top row) and for Github (bottom row). Compare with rLLCs in Figure 2.

For all the results in the paper, we used a value of 100 for the former and 50 for the latter. However, we have found little dependence of the results on the values of these hyperparameters, beyond the fact that lower values lead to more noisy estimates. This is in contrast to the LLC, where we have found that hyperparameter tuning is important for the final results.

We also compute the Hessian trace for subsets of parameters, and for different datasets, in a similar manner to the LLC. Weight refinement is performed by considering the loss as a function of only the selected parameters rather than all parameters, and taking the Hessian of that function. Data refinement is performed by using a different dataset for computing the loss. We measure the same parameter subsets and datasets as for the LLC (Section 4).

### D.1.2 RESULTS

The Hessian trace results are plotted in Figure 14, which we compare to the rLLC results in Figure 2. We immediately see both similarities and differences between these curves and the rLLCs, both supporting the validity of the information provided by the rLLCs while also providing tentative evidence that the rLLCs do not merely recapitulate the information contained the Hessian trace.

The unrefined, full-model Hessian trace does appear to detect the transition between LM1 and LM2 and between LM3 and LM4, but fails to distinguish between LM2 and LM3. Additionally, the Hessian and the LLC disagree on what happens after LM4 – from the LLC's perspective, the complexity of the model mostly stops increasing, whereas the Hessian trace continues to rise.

Zooming into specific components by weight-refining the Hessian shows a number of connections to the structural changes that were investigated earlier. The first notable similarity of the Hessian trace is its ability to (partially) distinguish the differentiation that different clusters of attention heads go through.

In layer 0, we see that the previous token heads, 0:1 and 0:4, extend their middle-of-training plateau region up until their composition scores with the induction heads are maximized. We also see the layer 0 multigram heads sharing a peak in value around the end of LM2 or LM3, which approximately matches the timing of a critical point in their K-composition scores (cf. Figures 7, 5). Finally, 0:5 is visually separated from the rest of the heads just by its magnitude, and the end of its plateau region approximately matches the time when it has its spike in K-composition scores (cf. Figure 7).

In layer 1, the clustering seems more ambiguous. Most of the multigram heads seem to have a similar shape, but it also seems plausible to include one of the induction heads, `1:6`, with the rest of the layer 1 multigram heads. The other induction head, `1:7`, is differentiated from other heads by a later peak.

Overall, we do see non-trival developmental signals from the weight-refined Hessian traces. However, there are also significant drawbacks, such as the apparent instability. As a result, it is clear where the critical points are for wrLLCs, but this is often not the case for the Hessian trace. For example, heads like `0:1`, `0:4`, and `0:5` appear to have noisy plateaus as opposed to obvious peaks. One might be tempted to adopt a heuristic that the *end* of a plateau is what matters, but this heuristic becomes problematic for heads like `1:6`. A priori, there is no way to distinguish the second peak of `1:6` between a real, distinct critical point, the end of a plateau starting from its first peak, and simply an artifact due to the instability of the trace.

We may also examine the data-refined Hessian trace, specifically on the GitHub dataset. This appears substantially harder to interpret than the Hessian trace on the original Pile dataset: the data is much more noisy, does not appear to respond to most developmental stage boundaries, and does not appear to add information beyond that of the original weight-refined Hessian traces. However, it does remain possible that this is a methodological issue — some earlier runs with less compute yielded less noisy data. Therefore, we believe the data-refined Hessian trace yields inconclusive results.

### D.1.3 DISCUSSION

Given that the Hessian trace appears to find developmental information related to that found by the LLC, it is worth comparing the two. Compared to the LLC, the Hessian trace has both advantages and disadvantages. The Hessian trace is more popular in the literature, and is easier to tune hyperparameters for. On the other hand, the Hessian trace has less theoretical support compared to measures like the Hessian rank or LLC, and (unlike the LLC) the Hessian trace cannot measure the behavior of the loss function beyond second order, by definition. Empirically, for the language model we studied, the LLC appears to present a clearer picture of the model's development than the Hessian trace.

As a separate point, it is worth emphasizing that the Hessian trace is measured in a substantially different manner to the LLC: the Hessian trace uses second order information at a single parameter, whereas LLC estimation uses first order information at many parameters near the original parameter. This decreases the likelihood of correlated mistakes between the two methods, and makes any agreement between them non-trivial.

### D.2 FIM TRACE

The Fisher information matrix (FIM) can be seen as a particular kind of Hessian matrix (Martens, 2020), making the FIM trace closely related to the Hessian trace discussed in Appendix D.1. Compared to the Hessian, the FIM has the advantage of always being positive semi-definite, as well as having deeper information-theoretic roots (Amari, 2016). We find that the results of the FIM trace are qualitatively similar to that of the Hessian trace, but with more noise due to the estimation methodology.

### D.2.1 METHODOLOGY

The Fisher information matrix (FIM) is given by

$$I_{jk} = \mathbb{E}_{x \sim p(x|w)} \left[ \left( \frac{\partial}{\partial w_j} \log p(x|w) \right) \left( \frac{\partial}{\partial w_k} \log p(x|w) \right) \right]$$

where $p(x|w)$ gives the model's probabilities over data samples $x$ given a parameter $w$ (Efron & Hastie, 2021). Implicitly, the use of $p(x|w)$ requires our model to be a probabilistic one; for models which are not immediately probabilistic, this step may require some interpretation.

For language models, we choose to interpret the data samples $x$ as strings of tokens of context-window length, and given such a string $x$ (for a fixed parameter $w$), the language model outputs the probability of this string, $p(x|w)$. Note that this differs from a more "literal" probabilistic interpretation of the model, where the model is treated as a supervised model $p(y|x, w)$, and yields next-token predictions over next-tokens $y$ given input previous-tokens $x$.

We prefer the former for two reasons: the $p(x|w)$ representation directly encodes the fact that a language model is really about natural language *sequences* (rather than next-tokens), and inference is more natural and direct from $p(x|w)$ (whereas inference from $p(y|x, w)$ requires multiple evaluations to string next tokens together).

In order to estimate the FIM trace, we first note that the FIM at a parameter $w^*$ coincides with the Hessian of the population loss at $w^*$ if the true distribution of the training data is given by $p(x|w^*)$, the model at $w^*$ (Martens, 2020). Via this relationship, we may reuse the methodology for Hessian trace estimation.

Then, in order to estimate the FIM trace, we need only replace the training data with samples from $p(x|w^*)$ (i.e., generated by the model itself), and repeat the Hessian trace methodology from Appendix D.1. For hyperparameters, we used 30 for the dataset sample count, and 5 for the Hutchinson sample count. All other methodology is identical to Appendix D.1.

Note as the parameter $w^*$ changes over training, new samples need to be drawn from $p(x|w^*)$ at each training checkpoint where we wish to estimate FIM trace — this introduces additional noise and computation cost into the estimation process.

It is possible to estimate the FIM trace in weight-refined manner similar to the LLC and Hessian trace (on the other hand, data refinement would need to be done differently), but we only estimate the FIM trace for the entire model.

### D.2.2   RESULTS

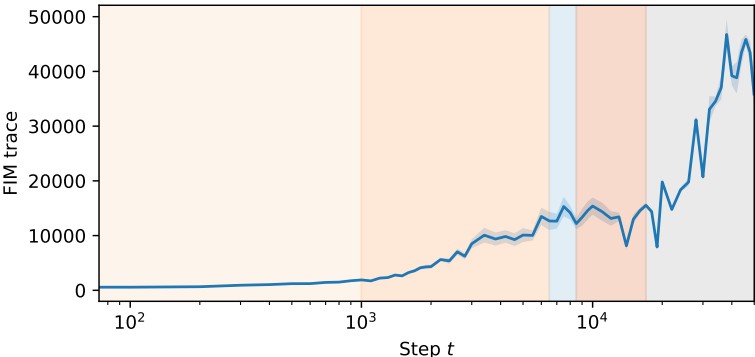

Figure 15: **The trace of the Fisher information matrix (FIM)** plotted over the course of training, with stages colored. Note the similarity to the Hessian trace from Figure 14.

The FIM trace is plotted over the course of training in Fig 15. The shape of the FIM trace appears to largely agree with the shape of the Hessian trace observed in Appendix D.1.

The extra noise is due to the fact that we must sample from $p(x|w^*)$ at each training step. This variance could possibly be reduced by e.g. keeping samples the same over all training steps, and just adjusting importance weights for the samples.

### D.2.3   DISCUSSION

Given that the FIM trace appears to largely match the Hessian trace, while being significantly noisier for a given amount of compute, we chose not to pursue it further.

### D.3   HESSIAN RANK

We also measure the (approximate) Hessian rank. From the perspective of singular learning theory, the Hessian rank is more principled than the Hessian trace, because it lower bounds the LLC[1].

---

[1]This can be concluded by applying the Thom splitting lemma to the population loss function $K(w)$, which changes variables so that $K(w)$ may be written a sum of a quadratic function $K_1(w_1) = w_1^T H w_1$ and an

However, empirically, we find that compared to the Hessian trace, our estimation of the Hessian rank is more computationally intensive, depends more on hyperparameter choice, and does not reflect structural development in the model as well. We consider these results inconclusive, as it is unclear if our methodology is measuring Hessian rank accurately; better estimation methodology could potentially avoid these problems.

### D.3.1 METHODOLOGY

The approximate rank of a square matrix $A$ can be defined as the number of eigenvalues of $A$ above some threshold $\tau$. Note the choice of this threshold can be a nontrivial hyperparameter in practice, and this was a significant difficulty for us.

Efficiently estimating the approximate rank of large matrices requires more sophisticated techniques and approximation than the typical algorithms used for smaller matrices. We use the technique described in Ubaru & Saad (2016), which we summarize here.

The rank estimation algorithm exploits the following facts:

- Spectral mapping theorem: if $p$ is a polynomial, $A$ is a square matrix, and $\lambda$ is an eigenvalue of $A$, then $p(\lambda)$ is an eigenvalue of $p(A)$.

- A step function is not a polynomial, but it may be approximated by one; in particular by Chebyshev polynomials, which are easy to compute efficiently.

- The trace of a matrix may be computed efficiently via matrix-vector products (see Appendix D.1), and if $p$ is a Chebyshev polynomial, the matrix-vector product $p(A)v$ is easy to compute efficiently.

- Combining all of the previous facts: if $A$ is a square matrix, and $p$ is a polynomial approximating a unit step function about a threshold $\tau$, then $\text{Tr}(p(A))$ efficiently yields approximately the number of eigenvalues above $\tau$ (the approximate rank with threshold $\tau$).

The details of this algorithm may be found in Ubaru & Saad (2016). As Hessian-vector products may be computed efficiently via autodifferentiation, we may use this algorithm to efficiently compute the approximate rank of the Hessian.

Because this method is equivalent to the Hessian trace estimator applied to $p(H)$, where $p$ is the polynomial from above and $H$ is the Hessian, the computational cost of this method is determined by the cost of computing the product $p(H)v$. Each one of these matrix-vector products requires some constant number of Hessian-vector products, determined by the degree of $p$, making the Hessian rank require a constant multiple more compute than the Hessian trace.

As this method relies on the trace estimation algorithm, it shares the two hyperparameters from that algorithm (the dataset sample count and Hutchinson sample count). It also has several additional hyperparameters: the degree of the polynomial $p$, the range of values for which we require $p$ to accurately approximate a step function, and the desired eigenvalue threshold $\tau$.

A higher degree for $p$ gives a more accurate approximation to the step function, allowing $p$ to rise faster and giving a more fine-grained rank approximation, but requires more compute. The larger the range we require the polynomial $p$ to be a good step function approximation, the slower $p$ will rise at the threshold value; however, the valid approximation range must at a minimum include the entire eigenvalue spectrum of the Hessian, or risk nonsensical results[2]. The threshold $\tau$ determines which eigenvalues are considered "zero" for the purposes of the approximate rank.

The degree of $p$ can be chosen based on required accuracy, or computational resources available. The range can be either set manually, or chosen automatically based on the min/max Hessian eigenvalues. The value for the threshold $\tau$ is harder to choose; there is often no clear eigenvalue gap or other indicator for where to set $\tau$, so we set it arbitrarily.

---

arbitrary function $K_2(w_2)$. Then Remark 7.2.3 from (Watanabe, 2009) tells us that the learning coefficient of $K(w)$ is equal to that of $K_1(w_1)$ plus that of $K_2(w_2)$, and because the learning coefficient of $K_1(w_1)$ is $\frac{r}{2}$ where $r$ is the rank of $H$, then the learning coefficient of $K(w)$ must be at least $\frac{r}{2}$.

[2]This is because the value of a Chebyshev polynomial outside its approximation range typically explodes to positive or negative infinity, so if any eigenvalue is outside the approximation range, the resulting trace also explodes.

There is another complication for the latter two hyperparameters: we are trying to estimate the Hessian rank *over the course of training*, not for a single training step. But the Hessian eigenvalue spectrum changes significantly over training; it is not clear if, or how, the approximation range and threshold should change over the course of training.

We try two strategies to set these hyperparameters, which we call the *fixed method* and the *adaptive method*.

**Fixed method**. Keep the approximation range and threshold the same over training. This means we are applying a consistent function to the eigenvalues over training, which is desirable. On the other hand, we need to set the approximation range large enough to deal with the largest eigenvalue seen over all of training ($\approx 5000$), even if most of the time the eigenvalues are much smaller. So we probably are not getting a good resolution estimate of the rank, especially early in training where the Hessian eigenvalues are smaller.

**Adaptive method**. Adjust the approximation range and threshold over training. We adapt the approximation range to the maximum and minimum eigenvalue at each training step, and set the threshold to a constant fraction of this range. This means we get better resolution when the eigenvalue spectrum is smaller, which is desirable. On the other hand, the function of the eigenvalues we measure will change at each training step, and this may confound the results.

For both strategies, we set the remaining hyperparameters to the same values: 3 for the dataset sample count, 1 for the Hutchinson sample count, and 100 for the degree of $p$. For the fixed method, we set the approximation range to (-1000, 6000), and the threshold $\tau$ to 500. For the adaptive method, we set the approximation range to $(-1.2|\lambda|, 1.2|\lambda|)$, and the threshold $\tau$ to $0.07\lambda$, where $|\lambda|$ is the largest eigenvalue of the Hessian in absolute value (estimated by power iteration).

While possible to measure weight-refined or data-refined Hessian rank, we did not measure this, given the difficulty we faced with estimating the Hessian rank for the full model on the original dataset.

### D.3.2 RESULTS

The Hessian rank over the course of training is plotted in Fig 16, for both hyperparameter selection methods discussed in the prior section. The fixed method appears to somewhat resemble the Hessian trace curve, but it barely changes early in training, and even when it does change later in training, it changes a relatively small amount in proportion to the overall value. The curve from the adaptive method has similarly low variance in comparison to its value, but in addition, does not appear interpretable.

### D.3.3 DISCUSSION

We believe these results are preliminary and inconclusive. Our assessment that this methodology is not producing good results stems from two main observations: (1) the failure to detect stage boundaries that are consistently identified by other methods (e.g., LLC, ED, Hessian trace, and behavioral metrics), and (2) the high sensitivity to estimation methodology, as evidenced by the discrepancy between fixed and adaptive methods and the dependence on numerous hyperparameters.

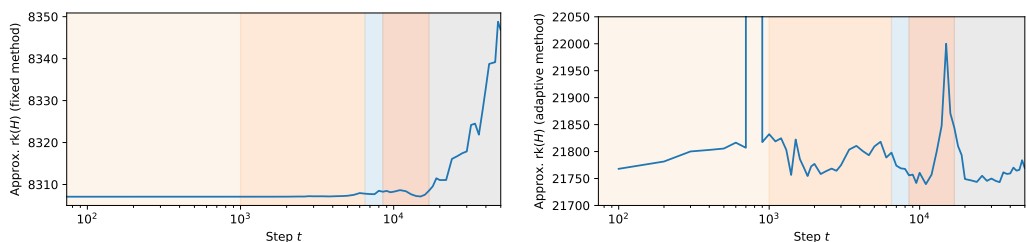

Figure 16: **The approximate rank of the Hessian** plotted over the course of training, estimated using the fixed method (left) and the adaptive method (right), with stages colored. Note y-axis scale in both charts: while both curves appear to show large changes in relative terms, they have little variation in absolute value.

These issues cast doubt on whether our estimated Hessian rank accurately captures the structural development in the model.

It remains unclear whether the problem lies in the theoretical underpinnings of the Hessian rank approach (e.g., its potential inability to capture higher-order degeneracy), in the practical implementation (due to flawed estimation methodology), or both. Given the apparent limitations of the current estimation methodology, we are unable to draw further conclusions at this time.

However, it does seem possible to improve the Hessian rank estimation methodology to deal with the issues we have highlighted. We suggest a few ideas for future work:

- To determine if the Hessian rank could even work well *in principle*, measure it in some toy model small enough to compute the Hessian eigenspectrum exactly (but still large enough to exhibit nontrivial stagewise development).

- Attempt something in between the fixed and adaptive methods: perhaps fixing the approximation range and threshold for each period of model training where the Hessian spectrum does not change significantly, or for each developmental stage.

- Fix noise in the adaptive method caused by noise in the max eigenvalue, by e.g. filtering the value of the max eigenvalue over time.

- Find a principled way to set the threshold $\tau$ based on some desired loss after dropping all eigenvalues below $\tau$. In turn, the desired loss could be set to a desired "effective compute" value via scaling laws.

# E COMPARISON AGAINST ABLATION ANALYSIS

## E.1 BACKGROUND

Ablation techniques are widely used in mechanistic interpretability to test hypotheses about the importance of specific components or activations in neural networks (Chan et al., 2022a; Wang et al., 2023; Rauker et al., 2023; Bereska & Gavves, 2024). By selectively removing or modifying parts of a model and observing the impact on performance, researchers aim to identify critical elements of the model's computation.

In this analysis, we compare three common ablation methods: zero ablation (Meyes et al., 2020; Hamblin et al., 2023; Nanda et al., 2023; Morcos et al., 2018; Zhou et al., 2018), mean ablations (Bereska & Gavves, 2024), and resampling ablations (Chan et al., 2022a; Hanna et al., 2023; Goldowsky-Dill et al., 2023; Lieberum et al., 2023). Our goal is to understand the relative strengths and weaknesses of each approach in comparison to the weight-refined LLC.

## E.2 METHODOLOGY

We compared the following ablation techniques:

- **Zero ablation**: Setting the targeted activations to zero.

- **Mean ablation**: Replacing the targeted activations with their mean value across the dataset.

- **Resampling ablation**: Replacing the targeted activations with those from a randomly selected different input (Chan et al., 2022a). In practice, we rolled the activations forward over the batch index (Chan et al., 2022a).

The *ablation score* is the difference between the loss before and after the ablation (computed over the same dataset over which we average the activations for the mean ablations).

Besides treating ablations as a baseline to compare rLLCs against in this section, we use ablations at several points throughout the rest of the paper (Figure 6 and Appendix B.1) to complement the other analyses. This requires introducing one additional technique:

- **Path patching** (Wang et al., 2023; Goldowsky-Dill et al., 2023): Path patching involves first running two forward passes: one on uncorrupted inputs/activations, and one involving a

corrupted inputs/ablations. Then, one runs a final forward pass, patching in the uncorrupted & corrupted activations so as to isolate the role of a specific computational path.

For example, for the path ablation in Figure 6, we mean-ablate the layer 0 multigram heads, then save the outputs of head `1:5`, then we run another (clean) forward pass and patch in `1:5`'s activations before the layer 1 readout layer. This lets us determine the particular influence of the layer 0 multigram heads on head `1:5`, without having to worry about layer 0 heads' other roles.

### E.3 RESULTS

Figure 17 displays the results of the different kinds of ablation over training time.

**Ablations differentiate heads.** As with the rLLC, we can use these ablation scores to distinguish several types of heads:

- **The previous-token heads** `0:1` and `0:4` can be identified by an increase in the ablation scores during stage LM4.

- **The current-token head** `0:5` is also distinguishable by its increase across the ablation scores starting towards the end of LM2 until it reaches a peak during LM4, after which it decreases. This is especially pronounced in the resampling ablation scores.

- **The induction head** `1:7` is clearly distinguished by the increase of the ablation scores in LM4. The other induction head `1:6` is less clearly distinct, though its ablation scores do have a different shape from the multigram heads.

- **The multigram head** `0:0` has similar rLLC curves to the other layer 0 multigram heads (though it is relatively larger, see Figure 2). The ablation scores suggest that `0:0` is a distinct type of head throughout much of training, with substantially higher values and a qualitatively distinct shape. It is not until LM5 that this heads ablation scores settle to a value comparable to the other layer 0 multigram heads. This complements the analysis in Appendix B.6 that suggests `0:0` starts out as a "space" head that attends to whitespace tokens.

The shape of ablation scores is mostly conserved across the different types of ablations, with one exception: `1:3`, which reaches a local maximum at the LM2-LM3 boundary for the zero ablation score that is not visible in the other ablation scores. There are other more minor differences for `0:5` (described above).

Intriguingly, the ordering of the final resampling ablation scores is very close to the ordering of the final rLLCs. These are both also similar to the ordering of the final Hessian traces (Figure 14).

### E.4 DISCUSSION

**Weights vs. activations.** The most pronounced difference between the wrLLC and the ablations considered here is that the wrLLC operates in weight space, while ablation methods work in activation space. In practice, the two approaches seem to be complementary. Indeed, the LLC can be interpreted as a kind of ablation: it measures the expected change in behavior (as measured by the loss) under "typical" perturbations to weights, where "typical" means the perturbation is drawn from the local posterior.

**Discrepancies between the wrLLC and ablations.** Both approaches identify key developmental stages and specialized heads, such as previous-token and induction heads. However, notable discrepancies exist:

- The ablation scores are better at identifying a discrepancy in `0:0`. On the other hand, the wrLLC is better at identifying that this head ultimately matures into multigram head (which we confirm separately).

- Some heads (e.g., induction head `1:6` and the current-token head `0:5`) are more distinguishable in the wrLLC than in ablations.

- Ablation methods are generally computationally more efficient. However, they lack the theoretical grounding of the wrLLC (which is reflected in the existence of many different possible choices of ablation).

Overall, the resampling ablation score most clearly differentiates the different kinds of heads, which we take as positive evidence in favor of the method (Chan et al., 2022a). However, the practical distinctions are small enough that we default to using mean ablations.

# F ADDITIONAL EXPERIMENTAL DETAILS

## F.1 TRAINING & MODEL DETAILS

**Model architecture:** We considered the same model architecture as in Hoogland et al. (2024); Olsson et al. (2022): a two-layer attention-only transformer, with the following specifications:

- Context length: 1024 tokens
- Residual stream dimension: 256
- Number of attention heads per layer: 8
- Layer normalization: Included
- Positional embedding: Learnable Shortformer-style

The resulting models contained approximately 3 million parameters. We implemented these models using the TransformerLens library (Nanda & Bloom, 2022).

For tokenization, we employed a modified version of the GPT-2 tokenizer, reducing the vocabulary size from 50,000 to 5,000 tokens. This adjustment allowed us to decrease the overall model size.

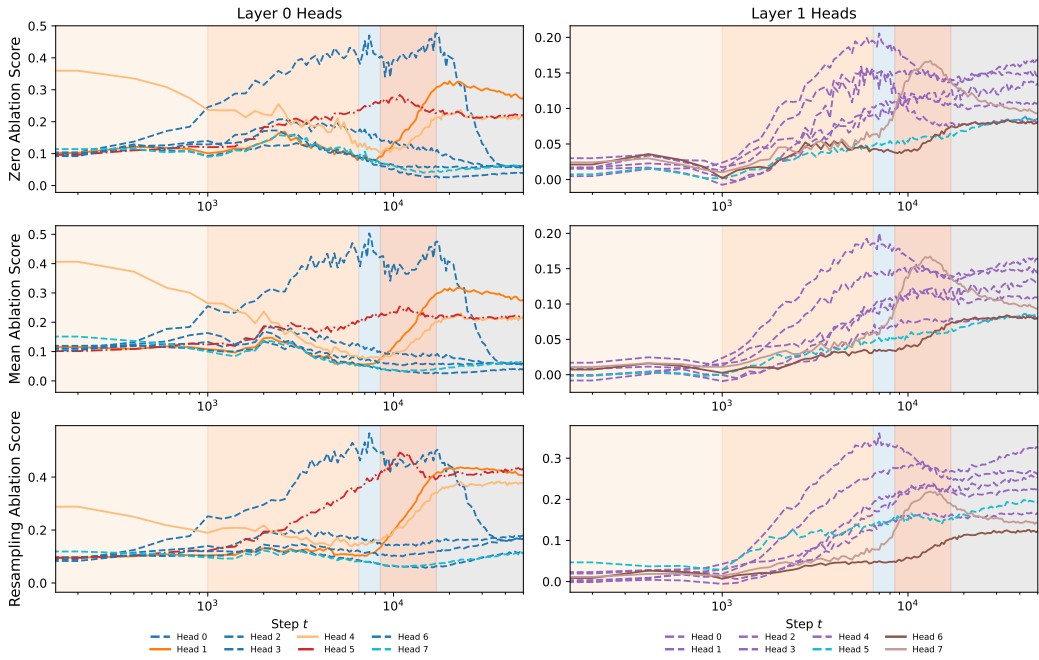

Figure 17: **Comparison of attention-head ablations over training time**. This figure shows the results of three different ablation techniques (zero ablation (top), mean ablation (middle), and resampling ablation (bottom)) applied to attention heads from layer 0 (left) and layer 1 (right). The y-axis shows the ablation score (difference in loss before and after ablation).

**Training:** The model and training run considered in the main body is the same as in Hoogland et al. (2024). This was trained on a subset of the DSIR-filtered Pile (Gao et al., 2020; Xie et al., 2023) for a total of around $50,000$ steps, with a batch size of 100.

## F.2 LLC ESTIMATION DETAILS

For estimating the Local Learning Coefficient (LLC), we employed Stochastic Gradient Langevin Dynamics (SGLD) with the following hyperparameters:

- The SGLD step size $\eta = 1e^{-3}$
- The inverse temperature $\beta = 30/n$
- The localization strength $\gamma = 200$
- Number of independent chains: 4
- Burn-in steps: 0
- Draws per chain: 200

The procedure is the same as described in Hoogland et al. (2024) using code from van Wingerden et al. (2024). The cost per SGLD-update is comparable to SGD with a linear increase in memory requirements.

We estimate that the computational cost of producing rLLC curves like those in Figure 1 for all heads scales like $CHT \log(N)$, where $C$ is a roughly constant O(100) number of SGLD samples, $H$ is the number of heads, $N$ is the total number of training steps, and $T$ is the cost of a single training pass.

## F.3 DATA-REFINED LLCS USING OTHER MODELS AS THE GENERATING PROCESS

In Section 4.3 we make use of a data-refined LLC where a particular trained one-layer attention-only transformer is used to provide the data distribution. For computational reasons, we make use of a modified form

$$\ell_n(w; M) = -\frac{1}{n} \sum_{i=1}^{n} \frac{1}{K-1} \sum_{k=1}^{K-1} D_{KL}\left(M(S_{\leq k}^i) || f_w(S_{\leq k}^i)\right) \tag{11}$$

of the empirical loss (1) where $M$ denotes the one-layer (L1) transformer.

**L0 and L1 models:** The L0 and L1 models used for these data-refined LLCs were trained analogously to the model considered in the main text and described in Appendix F.1

## F.4 COMPOSITION SCORES

Let $W_Q^h, W_K^h, W_V^h$ be the query, key, and value weights of attention head $h$ respectively. There are three types of composition between attention heads in transformer models in Elhage et al. (2021):

- Q-Composition: the query matrix $W_Q^h$ of an attention head reads in a subspace affected by a previous head
- K-Composition: the key matrix $W_K^h$ of an attention head reads in a subspace affected by a previous head
- V-Composition: the value matrix $W_V^h$ of an attention head reads in a subspace affected by a previous head

If $W_O^h$ is the output matrix of an attention head, then $W_{QK}^h = W_Q^{h\,T} W_K^h$ and $W_{OV}^h = W_O^h W_V^h$. The composition scores are

$$||MW_{OV}^{h_1}||_F / (||M||_F ||W_{OV}^{h_1}||_F) \tag{12}$$

Where $M = W_{QK}^{h_2\,T}$, $M = W_{QK}^{h_2}$, and $M = W_{OV}^{h_2}$ for Q-, K-, and V-Composition respectively.

### F.5 Clustering rLLCs

To better understand the patterns and relationships among the attention heads based on their rLLC trajectories, we applied various clustering algorithms to the rLLC data. This analysis aims to quantitatively group attention heads with similar developmental patterns.

#### F.5.1 Clustering Algorithms

We employed four different clustering algorithms to ensure a comprehensive analysis, using implementations provided by sklearn (Pedregosa et al., 2011) and tslearn (Tavenard et al., 2020):

1. **Euclidean K-means**: A centroid-based algorithm that partitions $n$ observations into $k$ clusters, minimizing the within-cluster sum of squares based on Euclidean distance (Macqueen, 1967).

2. **DTW K-means**: A variation of K-means that uses Dynamic Time Warping (DTW) as the distance measure, which is particularly suitable for time series data as it allows for non-linear alignment of sequences (Berndt & Clifford, 1994; Sakoe & Chiba, 1978).

3. **Hierarchical Agglomerative Clustering (HAC)**: A bottom-up approach that starts with each observation as a separate cluster and merges them iteratively based on a chosen linkage criterion (in this case the minimum variance criterion of Ward Jr 1963).

4. **Shape-Based K-means**: A custom implementation of K-means that uses Shape-Based Distance (SBD) as the distance measure. SBD is designed to capture similarities in the shape of time series, regardless of differences in scale or offset (Paparrizos, 2018).

In particular, we apply the clustering algorithms to the per-head Pile drLLCs and per-head Github drLLCs concatenated together. That is, the space in which we perform our clustering is $2T$-dimensional, where $T$ is the number of checkpoints. We found it unnecessary to apply standardization techniques to the trajectories before clustering as this led to marginal changes in the learned clustering.

#### F.5.2 Evaluation Metrics

To assess the quality of the clustering results, we used the following metrics:

1. **Silhouette Score**: Measures how similar an object is to its own cluster compared to other clusters. The score ranges from -1 to 1, where a higher score indicates better-defined clusters (Rousseeuw, 1987).

2. **Calinski-Harabasz (CH) Index**: Also known as the Variance Ratio Criterion, this index is the ratio of the sum of between-cluster dispersion and of within-cluster dispersion. A higher score indicates better-defined clusters (Caliński & Harabasz, 1974).

3. **Davies-Bouldin (DB) Index**: Calculates the average similarity between each cluster and its most similar cluster. A lower score indicates better clustering (Davies & Bouldin, 1979).

#### F.5.3 Results

**Clustering Layer 0 heads.** Figure 18 shows that as the number of clusters $c$ increases, the first distinction that emerges (for $c = 2$) within layer 0 is between the current-token head, previous-token heads, and multigram head `0:0` and the remaining multigram heads. Subsequently ($c = 3$), the first cluster splits apart: either the current-token head `0:5` or `0:0` breaks off from the rest, depending on the exact choice of clustering algorithm. The next distinction to emerge is between `0:7` and the bulk of the layer 0 attention heads.

As analyzed in Appendix B, heads `0:0` and `0:7` are, in fact, special kinds of multigram heads. Head `0:7` specializes to Dyck pattern (see Appendix B.3), and head `0:0` initially specializes to long skip-grams of spaces (see Appendix B.6; the exact function remains unknown).

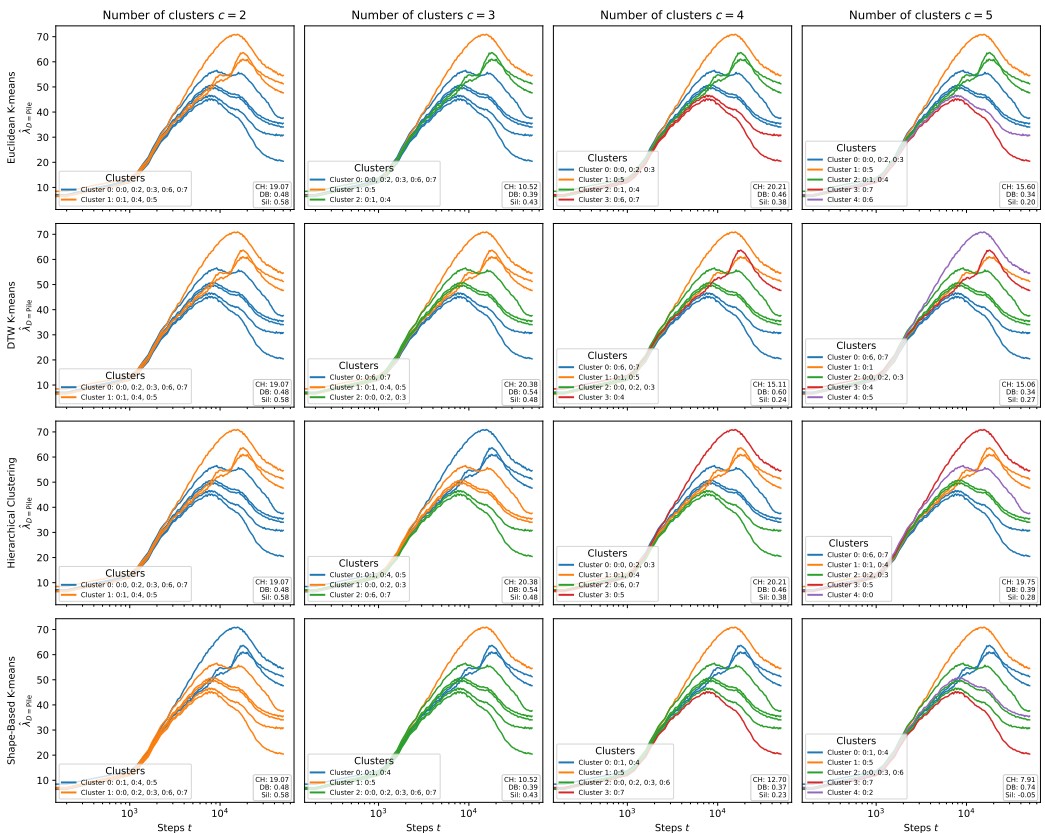

Figure 18: **Clustering results for Layer 0 rLLC trajectories** using various algorithms (rows) and number of clusters (columns). The x-axis represents training steps, and the y-axis shows the rLLC values. Each line represents an attention head, colored by its assigned cluster. As the number of clusters increases, we observe a clear separation between the current-token head, previous-token heads, and multigram heads. These clusters are fit on a concatenation of Pile drLLCs and Github drLLCs, but only the Pile drLLC component is displayed.

**Clustering Layer 1 heads.** Figure 19 shows that the clustering is extremely consistent within layer 1. The two induction heads are clearly distinguished from the bulk of the layer 1 heads. Upon increasing the number of clusters, the bulk of layer 1 multigram heads splits into pairs (see column 3 for $c = 4$ in Figure 19). Additionally, given enough clusters, the two induction heads become separated from one another.

As in the case of the layer 0 heads, these finer distinctions are meaningful: heads 1:5 and 1:3 are specialized to Dyck patterns (Appendix B.3), and the remaining heads are involved in more prosaic (skip) $n$-grams. This clustering is not perfect: in particular, clustering suggests a distinction between pairs 1:1/1:2 and 1:4/1:0, which is not obvious in the analysis by tokens in context (Appendix B).

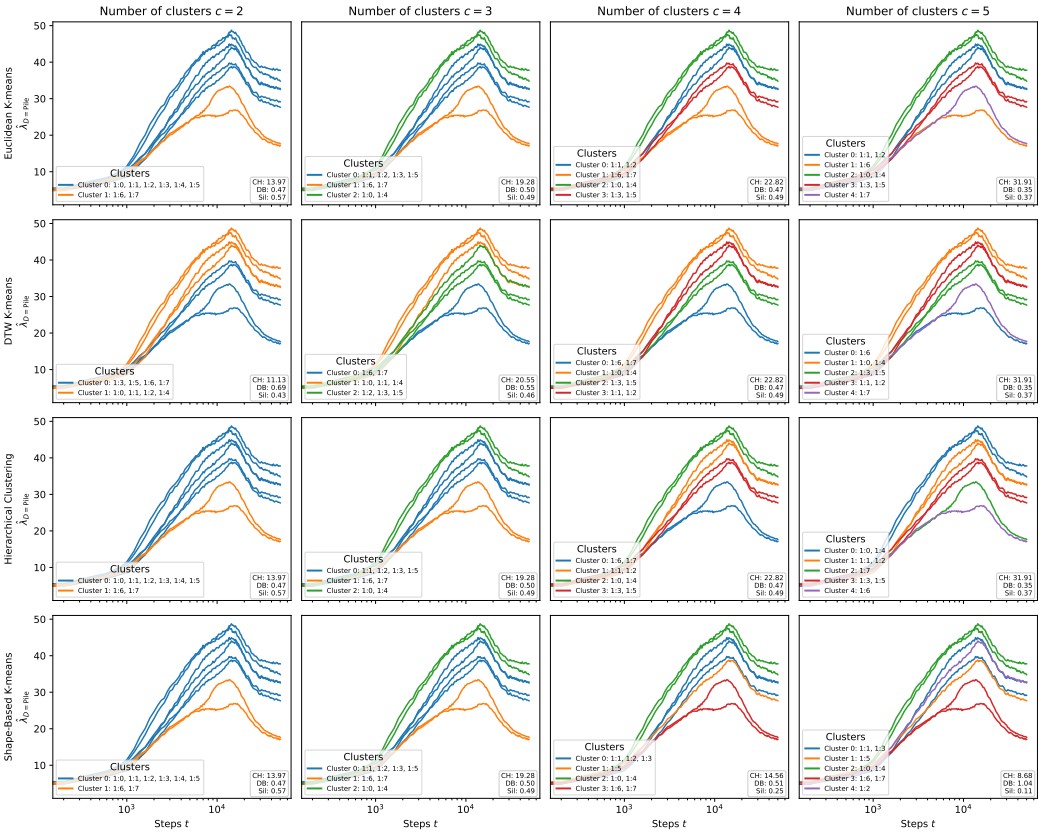

Figure 19: **Clustering results for Layer 1 rLLC trajectories** using various algorithms (rows) and number of clusters (columns). The x-axis represents training steps, and the y-axis shows the rLLC values. Each line represents an attention head, colored by its assigned cluster. The results consistently show a clear distinction between induction heads and other heads across all methods. These clusters are fit on a concatenation of Pile drLLCs and Github drLLCs, but only the Pile drLLC component is displayed.

We note that the clusters are largely unaffected when first standardizing the trajectories (by subtracting the mean and dividing by the standard deviation computed across steps and possibly heads).

**Generalization to other seeds.** In Appendix G, we show that these clusters appear to be universal across training runs: clusters fit to this seed generalize to different training runs.

### F.5.4 DISCUSSION

**Capturing shape similarities:** We chose to use the Euclidean K-means algorithm for our ultimate classification, with $c = 3, 4$ for layer 0 and $c = 2, 4$ for layer 1. We manually chose the number of clusters based on visual inspection.

Generally, the different clustering algorithms were in tight agreement. However, the Shape-Based K-means algorithm performed best when the clusters were fit to just the Pile drLLC data and not the Github drLLC data. This technique also transfers better to other seeds (Appendix G) and is closer to the results of clustering by eye. However, shape-based clustering performs worse when we increase the cluster count to attempt to resolve finer distinctions within the layer 1 multigram heads (Figure 1). In practice, we recommend taking a majority vote approach, with shape-based clustering getting additional weighting.

**Limitations of evaluation metrics:** It's important to note that the standard evaluation metrics (Silhouette Score, Calinski-Harabasz Index, and Davies-Bouldin Index) did not appear to work well

in our context. This is likely due to the low-sample setting of our analysis, with only 8 trajectories per layer, and due to the fact that Euclidean-distance-based diagnostic metrics may not be appropriate when the shape appears to be the more salient feature for clustering. These metrics are designed for larger datasets and may not provide reliable guidance in our case.

# G   COMPARISON WITH OTHER SEEDS

We trained three additional seeds identically to the original setup described in Appendix F and Hoogland et al. (2024). Applying weight- and data-refined LLCs to these training runs yields the figures in Figure 23.

The different training runs are very similar (Figure 23) but there are some differences:

1. **Some models develop only a single induction head and previous-token head**. Two of the runs have two heads each. Two of the runs have a single head each. Other than this, each run has a single current-token head, and the remainder of the heads seem to be multigram heads.

2. **In some runs, there is no LM3 stage** (in which the full-model LLC is decreasing). However, it is always the case that the multigram-head rLLCs begin decreasing after LM2.

Remarkably, despite these differences, clusters that are fit to the original seed generalize to per-head rLLCs *from different training runs*. Figure 20, Figure 21 and Figure 22 show the results of applying the shape-based clusters fit to seed 1 (Figure 18 and Figure 19, bottom rows) to different training runs. There are exceptions: e.g., for seed 2, layer 0 with $c = 3$ clusters, one of the previous-token heads is identified as a current-token head, and for seed 3 the current-token head does not get distinguished from the other heads by any crossing.

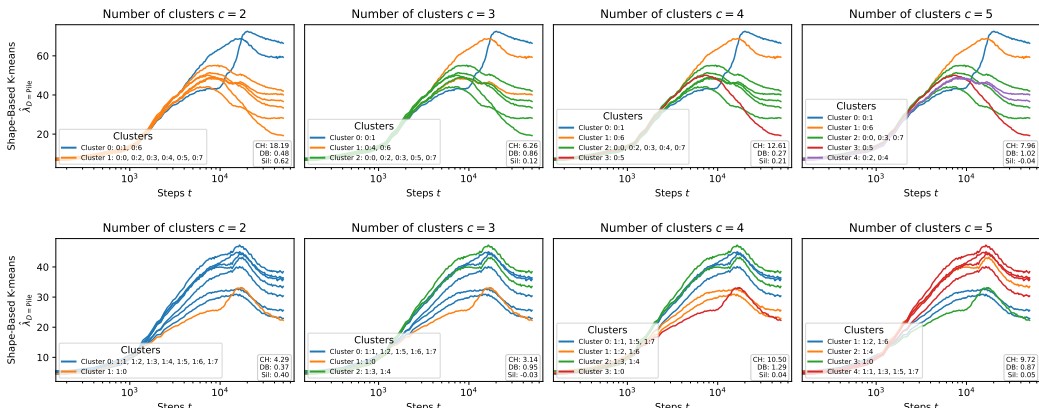

Figure 20: **Clustering validation on seed 2** for layer 0 (top row) and layer 1 (bottom row). Applying shape-based clusterings fit to the original seed Pile drLLC and Github drLLC generalize to held out seeds (Figure 18 and Figure 19, bottom rows). Here, the current-token head is 0:6, the previous-token head is 0:1, and the induction head is 1:0.

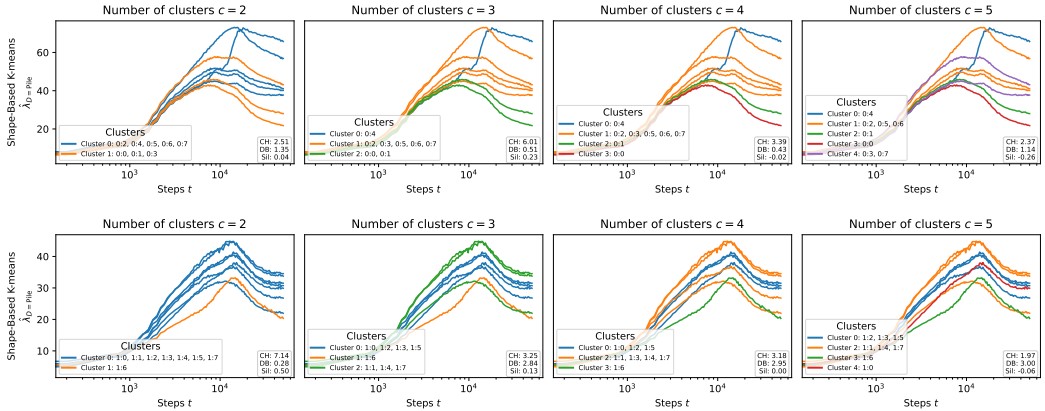

Figure 21: **Clustering validation on seed 3** for layer 0 (top row) and layer 1 (bottom row). Applying shape-based clusterings fit to the original seed Pile drLLC and Github drLLC generalize to held out seeds (Figure 18 and Figure 19, bottom rows). Here, the current-token head is $\boxed{0:2}$, the previous-token head is $\boxed{0:4}$, and the induction head is $\boxed{1:6}$.

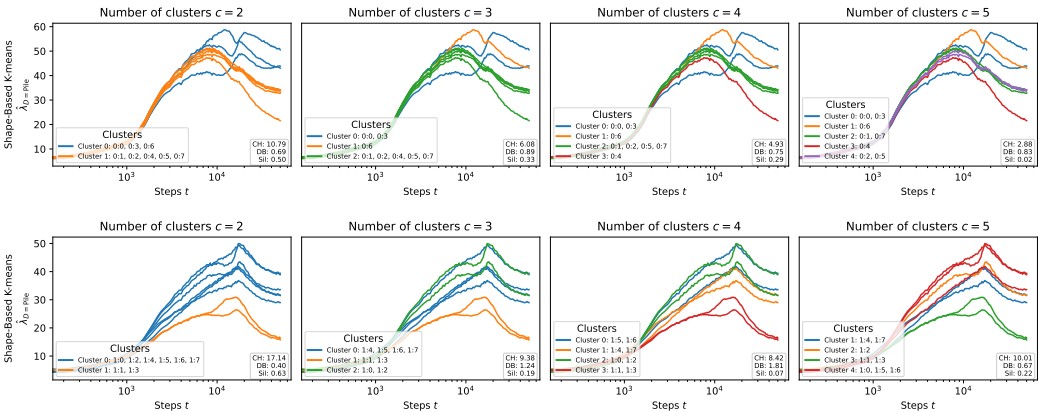

Figure 22: **Clustering validation on seed 4** for layer 0 (top row) and layer 1 (bottom row). Applying shape-based clusterings fit to the original seed Pile drLLC and Github drLLC generalize to held out seeds (Figure 18 and Figure 19, bottom rows). Here, the current-token head is $\boxed{0:6}$, the previous-token heads are $\boxed{0:0}$ and $\boxed{0:3}$, and the induction heads are $\boxed{1:1}$ and $\boxed{1:3}$.

## H COMPARISON WITH LARGER MODELS

In Figure 24, we show that the refined LLC generalizes to larger model settings. In particular, for Pythia-70m (Biderman et al., 2023), the Github wdrLLC (cf. Figure 3, right) distinguishes the previous-token heads and induction heads (that together make up the induction circuit) from other types of heads. To verify the identity of these heads, we use the previous-token and prefix-matching scores following Olsson et al. (2022).

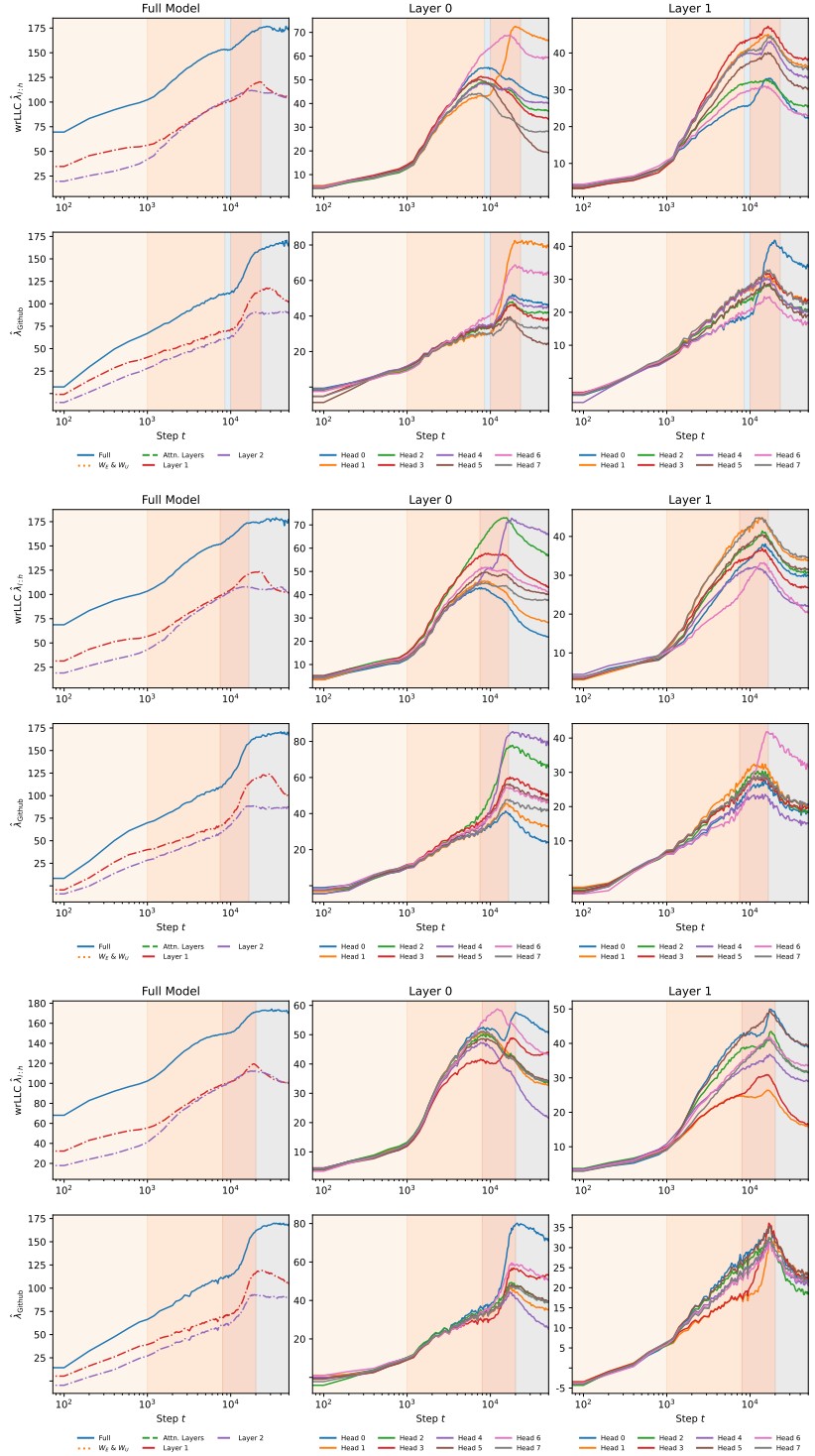

Figure 23: **Additional seeds show similar per-head rLLCs to the original training run.** Every seed has a current-token head. Every seed has at least one previous-token head and one induction head (the top two seeds only have a single one each). All of the remaining heads appear to be multigram heads.

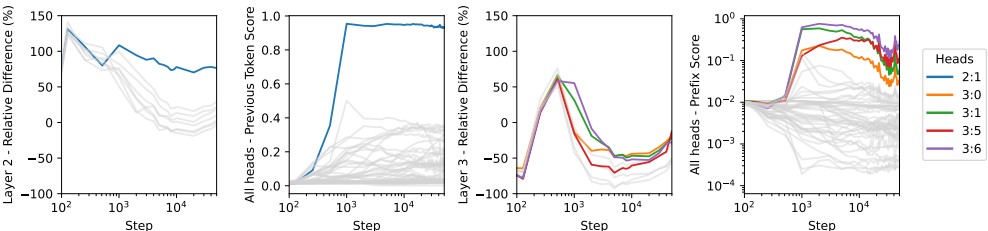

Figure 24: **The induction circuit in Pythia 70M revealed through data-refined LLCs**. The panels show different metrics tracking the development of the induction circuit: (Left) Relative difference between Pile and GitHub LLCs for Layer 2 heads with the outlier identified as the model's sole previous-token head. (Center-left) Previous-token scores across all heads confirm this identification. (Center-right) Relative LLC differences for Layer 3 heads reveal two clusters at the end of training: candidate induction heads and non-induction heads. (Right) Prefix scores confirms the cluster of layer 3 heads with higher relative rLLCs as the induction heads, with head 3:2 being an exception.

