# OpenReview forum: "Differentiation and Specialization of Attention Heads via the Refined Local Learning Coefficient"
_ICLR.cc/2025/Conference — ICLR 2025 Spotlight_

### Official Review · Reviewer_w7vo · 2024-11-02

**Soundness:** 3
**Presentation:** 3
**Contribution:** 3
**Rating:** 8
**Confidence:** 2

**Summary:**

This paper proposes two types of refined local learning coefficients (LLC), weight-refined LLC and data-refined LLC, and applies them to a two-layer attention only transformer to study the progressive differentiation and specialization of the attention heads. Both LLCs (defined in eq 5) are based on the estimated LLC (defined in eq 2), where suppressing q or V leads to weight-refined or data-refined LLC. The limitations of the proposed LLCs are given in Sec 3.3. Experimental results show that the refined LLCs reveal how attention heads differentiate (featured in Fig 2) and specialize (featured in Fig 3). In addition, combining the refined LLCs reveal internal structure related to multigram prediction that enables nested bracket-matching.

I am not an expert in this area, and therefore read the paper multiple times. I enjoyed reading it. My rating is 6, but I'm willing to raise the score if weakness #1 is addressed because it is preventing me from understanding the general applicability/feasibility of the proposed method.

[update during discussion] The authors have added experiments on a larger model, I therefore raised my rating.

**Strengths:**

- This paper is well-written, even a non-expert can easily follow the content if they are willing to spend some time figuring out the math.
- Although on a toy network, the results presented by this paper are strong and interesting.

**Weaknesses:**

- In Sec 6, the authors mentioned that *"The techniques pioneered in this paper for understanding internal structure in two-layer attention-only transformers can of course be applied to models at a larger scale or with different architecture."*. It would be more convincing if they present some results in the paper. [I'm willing to raise my rating if this can be addressed]
- [This is a suggestion] When I read the paper for the first time, I didn't immediately understand "what this paper unlocks" or "why this work is important" after finishing the introduction section. I suggest improving the writing of this section. For example, moving some text from Sec 6 to the introduction may be a good option.

**Questions:**

1. If the analysis is applied to transformers learned in different domains (e.g., decision transformer in RL), will we have similar observations?
2. Tiny issues in equations:
    1. In eq (1), if $f_w$ is a function from contexts to probability distributions, do you still need $\texttt{softmax}$?
    2. Is $\lambda$ in equation (5) missing a hat?
    3. Is the threshold $t$ in eq (4) and (5) the time steps? If not, consider changing it as $t$ is used in the figures as time steps.

---

> ### Author Response · Authors · 2024-11-18
>
> Thank you for the comments and clear indication of changes that would improve your score.
>
> **Larger scale**. We note that the underlying technique of LLC estimation (non-refined) has been validated in deep linear networks up to 100M parameters \[1\]. This means there is no technical obstacle to estimating refined LLCs at least at this scale, and the only current obstacles to scaling to much larger models is hyperparameter selection and computational cost, which are the subject of ongoing work. As mentioned in the top-level comment, we hope to include some preliminary results on Pythia models in a revision.
>
> **Importance**. The final paragraph of the introduction has been rewritten with this suggestion in mind, drawing a closer connection to the developmental perspective emphasised in the final section.
>
> **Questions:**
>
> 1. We are not sure (though we expect the same analysis to lead to similar observations in other settings). This would make for interesting follow-up work.
> 2. Equations:
>     1. No, if $f\_w$ mapped directly to probabilities, then you would not need a softmax.
>     2. This is actually an expression for the theoretical learning coefficient, so there is no need for a hat (see \[1\] for more).
>     3. Thank you for pointing this out. That is confusing\! We have changed it to \\epsilon in the revised version.
>
> \[1\] Z. Furman and E. Lau “Estimating the local learning coefficient at scale” 2024; as cited in the paper.

---

> ### Author Response · Authors · 2024-11-20
> **Generalization to larger models**
>
> Following your suggestion we tested our methodology on Pythia-70M and you can find the results in Appendix H. We hope this addresses your question about scaling to larger models.

---

### Official Review · Reviewer_YkMj · 2024-11-03

**Soundness:** 4
**Presentation:** 3
**Contribution:** 4
**Rating:** 8
**Confidence:** 4

**Summary:**

The authors introduce refined variants of the Local Learning Coefficient (LLC), a measure of model complexity grounded in singular learning theory, to study the development of internal structure in transformer language models during training. By applying
these refined LLCs (rLLCs) to individual components of a two-layer attention- only transformer, they were able gain novel insights into the progressive differentiation and specialization of attention heads. The methodology used in this paper reveals how attention heads differentiate into distinct functional roles over the course of training, analyzes the types of data these heads specialize to process, and discovers a previously unidentified multigram circuit. The authors conclude that their findings demonstrate that rLLCs provide a principled, quantitative toolkit for developmental interpretability,

**Strengths:**

The work is well grounded in the existing literature and theories. A good number of appropriate citations back this up that the authors build the work in this paper on.

The mathematical descriptions are brief and concise but sufficiently detailed and understandable. Appropriate level of detail.

Section 3.3 Limitations and the comparisons to related work within was much appreciated.

The degree and amount of supplementary information and detail in the appendices was excellent and greatly supports the main paper.

**Weaknesses:**

It would be beneficial to even if briefly define how the authors are explicitly using the terms 'component' and 'arbitrary data distribution' earlier on since it is the differentiating elements to other LLC metrics. The reader is left wondering and speculating as they continue to read in the way it is presented in the original paper.

The immediate next statement says "We focus mainly on the rLLCs of individual attention heads ...", implying diversity in how their new rLLC metric can be defined, tweaked, and/or applied.  But the authors have not even defined what it is. It is introducing a degree of variability prematurely that comes across as confusing. Before implying what they are 'focusing' on among several possibilities of (how?) their rLLC metric is defined, how about saying what it is first clearly and explicitly.

Similarly, the use of the term 'development' at the bottom of page 2 would be strengthened by giving the reader at this stage a brief one or two sentence understanding of the explicit meaning of the term as adopted in the paper. Italicizing it alone is not enough. The reader understands its important, but please help them in building understanding along the way.

**Questions:**

See comments above.

---

> ### Author Response · Authors · 2024-11-18
>
> We thank the reviewer for their careful reading of the paper and detailed comments.
>
> **Meaning of terms**. The description in the introduction has been changed to “which measures the complexity of a component of the model (for example an attention-head or whole layer) with respect to a given data distribution, which may differ from the training distribution. For example, we can measure the rLLC for a particular attention-head in a transformer trained on the Pile with respect to the distribution which conditions on a token sequence representing code.” We hope that clarifies also the sense in which the next paragraph refers to focusing on the special case where the component is an attention head.
>
> **Development on p.2**. Thanks for this suggestion, in the new revision, we have attempted to explain more clearly how we intend this term to be read and provide some more context for why we adopt the developmental perspective at the end of p.2.

---

### Official Review · Reviewer_qe7W · 2024-11-03

**Soundness:** 3
**Presentation:** 2
**Contribution:** 3
**Rating:** 6
**Confidence:** 2

**Summary:**

This paper introduces refined Local Learning Coefficients (rLLCs) as a method to analyze the internal structural differentiation within transformer models.
In particular, they introduce weight refined LLCs and data-refined LLCs.
By applying rLLCs to individual attention heads of a two-layer attention-only transformer model on different types of data, the authors aim to track how these components specialise and differentiate into distinct functional roles across training. The work provides insights into attention head specialisation based on the types of data processed, discovers specific structural patterns (e.g., multigram circuits), and highlights the correspondence between data distributional structure, loss landscape geometry, and model learning dynamics.

**Strengths:**

The paper presents a novel metric to track structure during training based on data structure and tracking the evolution and specialisation of attention heads, overcoming some of the limitations of previous metrics. The paper presents some novel insights into how heads specialise and the effect of data.
- The presented metric (rLLC) overcomes some of the existing limitations of the LLC method, including the assumption that the $w^*$ is a local minima (which during training, it is very unlikely to be)
- rLLC seems to in practice be able to differentiate attention heads by their specialisation, it is interesting to observe the higher complexity of heads performing memorisation and n-gram heads vs the induction ones (Figure 1)
- The authors show how with a different data (Github) the head specialisation changes and more importance is given to the induction heads (Figure 3)
- The experiments are well-documented, with comprehensive comparisons against other interpretability measures like Hessian and Fisher Information Matrix traces, adding robustness to the findings.

**Weaknesses:**

- the experiments limit themselves to a two layer transformer, it would have been interesting to see how the proposed rLLC metric would behave in a bigger transformer. Is it still trackable? Does it lead to a meaningful interpretation of results?
- what about different architectures? Line 465 in the related section mentions AlexNet, it would have been interesting to carry out a similar analysis on layers of AlexNet

**Questions:**

- it was not clear to me how the K-means clustering was obtained to generate the colours for the different lines in e.g. Plot 1. What is the K-means over?
- Have you considered how the rLLCs would scale with deeper, more complex transformer models? Are there any preliminary results or hypotheses about the performance and practicality of this method on architectures with more layers or MLP components?
- Beyond interpretability, have you explored any practical use cases where rLLCs could directly influence model design or training strategies (e.g. model pruning)?

---

> ### Author Response · Authors · 2024-11-18
>
> We appreciate the comments and suggestions for improvement.
>
> **Bigger transformers with more layers**: the rLLC can be measured for transformers at any scale, albeit with more computational cost and the hyperparameter selection process may become more difficult at larger scale. The inclusion of MLP layers or deeper networks does not present a serious obstacle. In this paper we concentrate on smaller models, where it is easier to have a comprehensive understanding of the ground truth of the model’s behaviour, in order to have something to compare our rLLC measurements to. We expect the same techniques to apply to larger models, and as mentioned in the top-level comment, hope to include some preliminary results in a revision during the discussion period.
>
> **Different architectures**: the rLLC methodology can indeed be straightforwardly extended to study the development of other architectural components, including convolutional layers. While the current paper focuses on attention heads to maintain a clear scope and narrative, we agree this would be interesting future work.
>
> **K-means clustering**: The K-means clustering was performed on the rLLC trajectories themselves, treating each head's concatenated Pile and Github rLLC curves as a point in a 2T-dimensional space (where T is the number of checkpoints). We found that using both datasets' trajectories helped better distinguish the functional types of heads (using individual datasets also works, but leads to more varied outcomes over different clustering algorithms). For transparency, we have included detailed clustering methodology in Appendix C.2, including comparisons with other clustering approaches like DTW K-means and Shape-Based K-means that are specifically designed for time-series data. The clusters are robust across these different methods and also generalize to other training runs (Appendix F).
>
> **Beyond interpretability**: we haven’t considered any of these yet, but the connection to model pruning seems particularly promising to us.

---

### Official Review · Reviewer_RBF6 · 2024-11-04

**Soundness:** 3
**Presentation:** 4
**Contribution:** 3
**Rating:** 6
**Confidence:** 3

**Summary:**

The authors introduce a novel approach to understanding the internal specialization of attention heads in Transformer models through Refined Local Learning Coefficients (rLLCs). By measuring the complexity of attention heads and tracking their developmental process during training, the authors reveal how different heads gradually specialize in performing specific tasks.

**Strengths:**

1. The rLLC provides an innovative, quantitative tool for tracking specialization in Transformer models, which could have applications beyond this study.

2. The paper includes a comprehensive analysis of attention head specialization, backed by clustering and ablation studies that demonstrate the functional roles of different heads.

3. The discovery of "multi-phrase circuits" and the evidence of cross-layer head collaboration provide valuable insights into how Transformer models handle complex linguistic patterns.

**Weaknesses:**

1. While the method is effective on smaller models, the paper does not discuss the computational implications for larger models.

2. The study focuses on understanding Transformer models without discussing how these insights could guide architecture optimization or training strategies in practice.

**Questions:**

1. Could insights from rLLC tracking be used to optimize training processes, such as by dynamically adjusting learning rates or selectively freezing layers based on head specialization patterns?

2. How well does the rLLC methodology generalize to larger models, such as GPT-like architectures?

3. How do rLLCs compare with the existing interpretability metrics? The authors could refer to these related works: [Michaud, Eric, et al. "The quantization model of neural scaling." Advances in Neural Information Processing Systems 36 (2024).] ; [Xiao, Xiongye, et al. "Exploring neuron interactions and emergence in llms: From the multifractal analysis perspective." arXiv preprint arXiv:2402.09099 (2024).]

---

> ### Author Response · Authors · 2024-11-18
>
> Thank you for your comments and questions.
>
> **Larger models.** Here are some details on the computational cost for running these experiments, which we have added to the revision (appendix F.2): running an SGLD chain is approximately as expensive as running SGD training for the same number of steps. We estimate that the cost of producing rLLC curves like those in Fig 1 for all heads scales like C\*H\*log(N)\*T, where C is a roughly constant O(100) number of SGLD samples, H is the number of heads, N is the total number of training steps, and T is the cost of a single training pass.
>
> There is no essential difference between a “GPT-like” architecture and our models, except for scale and the inclusion of MLP layers. LLC estimation for transformer models with MLP layers was done in \[1\] and we don’t expect any fundamental obstacle in performing rLLC estimates for MLP layers just as we have done in the present paper for attention heads. As mentioned in our top-level comment, we will share preliminary results for larger Pythia models in a coming revision.
>
> **Architecture optimization and training processes**. We acknowledge that the focus of our paper is scientific understanding, and we have not thought about how these results might be used to inform the design of better architectures or training processes.
>
> **Comparison with existing interpretability metrics**. Thank you for the references. To the best of our knowledge Michaud et al. does not introduce any interpretability metrics. Xiao et al seems very interesting, and we thank the referee for bringing it to our attention. We prioritised comparing the rLLC against two techniques commonly in use in the literature, namely Hessian-based complexity metrics and ablation interpretability analysis.
>
> In Appendix D, we compared the rLLC as a complexity metric against the Hessian trace, the Fisher Information Matrix (FIM) Trace, and the Hessian Rank. We found that “the \[r\]LLC appears to present a clearer picture of the model’s development than the Hessian trace” and that the FIM Trace looks similar to the Hessian trace but noisier. Our Hessian Rank results were inconclusive due to difficulties in estimation.
>
> In Appendix E, we compared the use of the rLLC as an interpretability tool against zero, mean, and resampling ablations, as well as path-patching. We found that these four ablation techniques taken together allow distinguishing heads and stages to a similar extent to the rLLC, with some heads more easily distinguished using the rLLC, and some more easily using ablations. Note, however, that these two techniques are complementary, as the rLLC operates in weight space and ablations in activation space. Finally, we want to reiterate that ablations are somewhat more computationally efficient, yet are less theoretically grounded (reflected in the fact that one has to choose which ablation to run).
>
> \[1\] Hoogland et al, 2024; cited as in the paper.

---

### Author Response · Authors · 2024-11-18

Thank you for reading our paper and your suggestions for improvement. Our detailed replies are below. There was one common theme that we would like to address in this top-level comment:

**Model scale**. Several reviewers asked about the implications for larger models, and whether the techniques would generalize. In principle there is no obstacle to scaling these techniques to larger models: indeed, the underlying theory (singular learning theory) is not specific to any particular model scale, and we expect that SGLD-based LLC estimation should also scale. Our primary reason for focusing on small transformers is that it is cheaper and faster to iterate at this scale, and these were important advantages for the present work. Now that we have exhaustively validated the use of refined LLCs in these small transformers, we are confident that it is worthwhile to spend the time and resources to examine larger models, which we plan to pursue in future work.

However, in order to address the reasonable concerns of reviewers about scalability, we plan to include some preliminary results on refined LLCs for Pythia models in a revision later this week.

---

### Author Response · Authors · 2024-11-20
**Generalization to larger models**

We have added Appendix H demonstrating that the rLLC methodology generalizes to larger models. Specifically, we apply the technique to Pythia-70M and show that the relative difference between Pile and GitHub data-refined LLCs (following the procedure from Section 4.2 and Figure 3) successfully distinguishes previous-token heads and induction heads from other attention heads. This validates that our approach for identifying specialized heads through data-refined LLCs extends beyond the two-layer attention-only transformer studied in the main text.

---

### Meta-Review · Area_Chair_QCYq · 2024-12-20

**Metareview:**

This paper investigates how attention heads specialize throughout the training of a 2-layer Transformer, and how this specialization is correlated with the "local learning coefficient" (LLC) around that attention head. More specifically:
  * The LLC roughly corresponds to how much a single weight affects the overall loss, and thus can be used to track the weights associated to every attention head throughout training, to obtain an LLC curve for every head.
    * The LLC curves are then clustered together following how similar their shapes are.
  * Independently, the paper also clusters attention heads according to their specialized function ("N-gram", "skip-gram", etc.)
    * This is done by manually inspecting and visualizing attention scores between pairs of tokens.
  * Experimentally, it turns out that the LLC clusters and the attention specialization clusters are highly correlated, which means that LLC metrics can be used as more automated metrics for checking attention specialization.

## Strengths
* This paper is very well and professionally written, with clear language, figures, and the right balance of mathematical detail between the main body and the appendix.
* There's a lot of additional high quality ablations found in the Appendix, making this work very comprehensive in its investigation. In addition, there were multiple explanations for unfamiliar readers, around concepts such as "Dyck Head", "Multigrams", etc. which was very useful in allowing a detailed read.

## Weaknesses
* There's a lot of machinery in this paper, although the authors did their best in concisely explaining the basics of their tools. But I still needed to re-read the paper and even the Appendix multiple times, to get the gist of:
  * Appendix B: How an attention head was classified as a "Dyck Head", "Induction Head", etc.
  * Appendix A + Equation 5: The intuitive meaning of weight + data refined LLC, and why this is computed as log of the volume ratio (which can be seen as some form of dimensionality?)
* Although the paper cites previous literature to motivate the construction of the LLC metric, at the end of the day, it's tracking "how much a weight affects the loss". It's not clear whether very specific definitions are needed, or they can be simplified. For example:
  * Why is the Gibbs Posterior needed for the distribution? Couldn't we just uniformly sample from a L2-ball?

**Additional Comments On Reviewer Discussion:**

The reviewers unanimously agree that this paper should be accepted, with high post-rebuttal scores of (6,6,8,8), which definitely puts it in the spotlight category.

But it's unclear whether the paper should be moved to oral presentation, due to some of its esoteric math and machinery. This is evidenced by:
* All reviewers with the exception of YkMj, submitting lower confidence scores, signaling they're not familiar with some parts of the work and couldn't check some of the central parts.
* My meta-review and around the mathematical formulation of the LLC (granted, I'm also not an expert within this area).

Thus it's possible that this paper could be moved up to oral, but unfortunately it's too late to add a reviewer with much more experience within this field. My confidence score also reflects this.

---

### Decision · Program_Chairs · 2025-01-22

Accept (Spotlight)